# TRACTABLE MCMC FOR PRIVATE LEARNING WITH PURE AND GAUSSIAN DIFFERENTIAL PRIVACY

**Yingyu Lin**[1*]**, Yi-An Ma**[1*]**, Yu-Xiang Wang**[2*]**, Rachel Redberg**[2]**, Zhiqi Bu**[3]
[1]UC San Diego, [2]UC Santa Barbara, [3]Amazon AI
`{yil208,yianma}@ucsd.edu {yuxiangw,rredberg}@cs.ucsb.edu zhiqibu@amazon.com`

## ABSTRACT

Posterior sampling, i.e., exponential mechanism to sample from the posterior distribution, provides $\varepsilon$-*pure* differential privacy (DP) guarantees and does not suffer from potentially unbounded privacy breach introduced by $(\varepsilon, \delta)$-approximate DP. In practice, however, one needs to apply approximate sampling methods such as Markov chain Monte Carlo (MCMC), thus re-introducing the unappealing $\delta$-approximation error into the privacy guarantees. To bridge this gap, we propose the Approximate SAmple Perturbation (abbr. ASAP) algorithm which perturbs an MCMC sample with noise proportional to its Wasserstein-infinity ($W_\infty$) distance from a reference distribution that satisfies *pure* DP or *pure* Gaussian DP (i.e., $\delta = 0$). We then leverage the Metropolis-Hastings algorithm to generate the sample and prove that the algorithm converges in $W_\infty$ distance. We show that by combining our new techniques with a localization step, we obtain the first nearly linear-time algorithm that achieves the optimal rates in the DP-ERM problem with strongly convex and smooth losses.

## 1 INTRODUCTION

The strongest form of differential privacy (DP) is $\varepsilon$-*pure* DP, which provides an inviolable bound of $\varepsilon \geq 0$ on an algorithm's privacy loss. The posterior sampling mechanism associated with the loss functions (Wang et al., 2015; Dimitrakakis et al., 2017; Gopi et al., 2022) is known to achieve near-optimal privacy-utility tradeoff for learning under pure DP — in theory. In practice, sampling from a general posterior distribution is typically either intractable or inefficient. Both issues can be solved by approximating the posterior, e.g. via MCMC sampling. But of course there is no free lunch: the approximation error introduced by this approach now degrades the privacy guarantee to the weaker notion of $(\varepsilon, \delta)$-*approximate* DP, under which we risk a catastrophic privacy breach with some small probability $\delta$. Our paper bridges the gap between theory and practice — thus alleviating a common growing pain of DP research — by proposing an efficient MCMC-based algorithm that samples from an approximate posterior *while satisfying pure DP*.

Recent research has explored various implementations of posterior sampling mechanisms, facing a recurring challenge: each approach either compromises on DP guarantees or incurs substantial computational costs. Studies by Wang et al. (2015); Gopi et al. (2022) illustrate that employing *approximate* posterior sampling with sampling error solely in terms of the total variation (TV) distance downgrades the DP guarantee to approximate DP. In contrast, Seeman et al. (2021) samples from the *exact* posterior with pure DP. But the associated runtime can be exponentially large due to a rejection sampling scheme that, in the worst-case scenario, has an exponentially low acceptance rate. This motivates the following question:

> Can we obtain *pure* DP guarantees with an efficient MCMC algorithm?

In this paper, we answer the question in the affirmative by developing the Approximate Sample Perturbation (ASAP) algorithm.

---

[*]Equal contribution.

## 1.1 OUR CONTRIBUTIONS

The main results of our paper are threefold.

1. We propose Approximate Sample Perturbation (Algorithm 2), a novel MCMC-type method designed to maintain pure DP and pure Gaussian DP by perturbing an MCMC sample. Theorem 2 demonstrates that ASAP maintains these DP guarantees when the MCMC sample distribution closely approximates the exact posterior, measured in terms of the Wasserstein-infinity ($W_\infty$) distance.

2. We establish a novel generic lemma (Lemma 8) that facilitates the conversion of TV distance bounds to $W_\infty$ distance bounds. This transformation enables the maintenance of pure DP guarantees when employing MCMC samplers with TV distance errors. Notably, this lemma is of independent interest.

3. We introduce the Metropolis adjusted Langevin algorithm (MALA) with constraint (Algorithm 1) that converges with respect to the $W_\infty$ distance. Integrated with a preceding *localization* step, detailed in Algorithm 3, it empowers the ASAP framework to achieve optimal rates in nearly linear time while ensuring pure DP and pure Gaussian DP in the context of the Differential Privacy - Empirical Risk Minimization (DP-ERM), for strongly convex and smooth losses.

## 1.2 RELATED WORK

Posterior sampling mechanism, i.e., outputting a sample $\hat{\theta} \sim p(\theta) \propto \exp(-\gamma(F(\theta) + \frac{1}{2}\lambda\|\theta\|_2^2))$ is a popular method for DP-ERM and private Bayesian learning. It was initially analyzed under pure DP (Mir, 2013; Dimitrakakis et al., 2017; Wang et al., 2015), then later under approximate DP (Minami et al., 2016), Rényi DP (Geumlek et al., 2017) and Gaussian DP (Gopi et al., 2022). In each case, it is known that with appropriate choices of $\gamma, \lambda$ parameters, it achieves the optimal rate (Bassily et al., 2014) under each privacy definition. However, the computational complexity of the MCMC methods has been a major challenge for this problem. To the best of our knowledge, Bassily et al. (2014); Chourasia et al. (2021); Ryffel et al. (2022); Mangoubi & Vishnoi (2022) are the only known results that obtain DP guarantees using MCMC methods without having $\delta > 0$. Bassily et al. (2014)'s sampler (a variant of Applegate & Kannan (1991)) runs in $O(n^4)$ and requires explicit discretization. Chourasia et al. (2021); Ryffel et al. (2022)'s algorithms run in nearly linear time (for strongly convex DP-ERM) and enjoy pure Rényi DP, but their utility bound is a factor of condition number $\kappa$ worse than the statistical limit. Mangoubi & Vishnoi (2022)'s algorithm translates the TV distance guarantee to infinity-distance ($d_\infty$) guarantee by incorporating uniform noise and a membership oracle, while their focus is on Lipschitz continuous loss. The idea of perturbing distributions to strengthen the DP guarantee is discussed in Feldman et al. (2018) and its application to Langevin analysis (Altschuler & Talwar, 2023), where additional noise converts $W_\infty$ guarantee to Rényi DP guarantee. Our work is the first that archives the optimal rate for Strongly-Convex DP-ERM under pure DP and pure Gaussian DP with a nearly linear time algorithm (for $\kappa = \text{polylog}(n)$).

DP-ERM can also be solved using other methods that do not require MCMC sampling. However, these methods either do not achieve optimal rates (output perturbation (Chaudhuri et al., 2011)) or are computationally less efficient (Noisy SGD (Bassily et al., 2014; Abadi et al., 2016)) or require the model to be (generalized) linear (e.g., objective perturbation (Chaudhuri et al., 2011; Kifer et al., 2012; Redberg et al., 2023)).

## 2 PROBLEM SETUP AND PRELIMINARIES

**Symbols and notations.** Let $\mathcal{X}$ be the space of data points, $\mathcal{X}^* := \cup_{n=0}^{\infty} \mathcal{X}^n$ be the data space, and $D \in \mathcal{X}^*$ be a dataset with an unspecified number of data points. Let the parameter space $\mathcal{U} \subseteq \mathbb{R}^d$ and for each $x \in \mathcal{X}$ and $\theta \in \mathcal{U}$, $\ell_x(\theta)$ denotes the loss function (or negative log-likelihood function). When $D = \{x_1, ..., x_n\}$, we denote $\ell_{x_i}(\theta)$ by $\ell_i(\theta)$ as a short hand. Denote the total loss $\mathcal{L} = \sum_{i=1}^{n} \ell_i$. For any set $S$, we denote the set of all probability distributions as $\Delta^S$ or $\mathcal{P}_S$, so that a mechanism $\mathcal{M} : \mathcal{X}^* \to \Delta^{\mathcal{U}}$ is a randomized algorithm. We use $\mathcal{M}(D)$ to denote the probability distribution as well as the corresponding random variable returned by the mechanism. For a set $S$, we denote $\text{Diam}(S) := \sup_{x,y \in S} \|x - y\|_2$, and $\|S\| := \sup_{x \in S} \|x\|_2$.

## 2.1 DIFFERENTIAL PRIVACY EMPIRICAL RISK MINIMIZATION (DP-ERM)

Empirical risk minimization (ERM) is a classic learning framework which casts the problem of finding a "good" model into an optimization task. Our goal is to find the parameter $\theta^*$ in the parameter space $\mathcal{U} \subseteq \mathbb{R}^d$ which minimizes the empirical risk: $\theta^* = \arg\min_{\theta \in \mathcal{U}} \left( \sum_{i=1}^n \ell_i(\theta) \right)$.

The problem setting of our paper is differentially private empirical risk minimization (DP-ERM): empirical risk minimization under a privacy constraint.

## 2.2 DIFFERENTIAL PRIVACY DEFINITIONS

**Definition 1** (Differential privacy (Dwork et al., 2006; 2014)). *Mechanism $\mathcal{M}$ satisfies $(\varepsilon, \delta)$-DP if for all neighboring datasets $D \simeq D'$ and for any measurable set $S \subseteq Range(\mathcal{M})$,*

$$\mathbb{P}[\mathcal{M}(D) \in S] \leq e^\varepsilon \mathbb{P}[\mathcal{M}(D') \in S] + \delta.$$

*When $\delta = 0$, $\mathcal{M}$ satisfies $\varepsilon$-(pure) DP.*

Differential privacy (DP) provably bounds the privacy loss of an algorithm. *Approximate* DP ($\delta > 0$) is a practical and popular DP variant which allows a "failure" event — where the privacy loss exceeds the bound — to occur with some probability. Beware: approximate DP does not bound the *severity* of a privacy breach. The privacy loss under a failure event could be arbitrarily large. Avoiding this risk requires *pure* DP ($\delta = 0$), which provides a deterministic bound on the privacy loss.

In comparison to approximate DP, *Gaussian* DP is a more "controlled" relaxation of pure DP which does not have an unbounded failure mode. We can define Gaussian DP via the hockey-stick divergence.

**Definition 2** (Hockey-Stick Divergence). *The Hockey-Stick Divergence of distributions $P, Q$ is defined as*

$$H_\alpha(P\|Q) := \mathbb{E}_{o \sim Q}\left[ \left( \tfrac{dP}{dQ}(o) - \alpha \right)_+ \right]$$

*where $(\cdot)_+ := \max\{0, \cdot\}$ and $\frac{dP}{dQ}$ denotes the Radon-Nikodym derivative.*

**Definition 3** (Gaussian Differential Privacy (Dong et al., 2022)). *We say that a mechanism $\mathcal{M}$ satisfies $\mu$-Gaussian differential privacy if for any neighboring dataset $D, D'$,*

$$H_{e^\varepsilon}(\mathcal{M}(D)\|\mathcal{M}(D')) \leq H_{e^\varepsilon}(\mathcal{N}(0,1)\|\mathcal{N}(\mu,1)) \qquad \forall \varepsilon \in \mathbb{R}.$$

This definition is equivalent to the *dual* definition from the hypothesis testing perspective in (Dong et al., 2022, Definition 2.6). The statement naturally provides a "dominating pair" of distributions, facilitating adaptive composition, amplification by sampling, and efficient numerical computation (Zhu et al., 2022).

## 2.3 EXACT POSTERIOR SAMPLING: DP AND UTILITY GUARANTEES

The primary algorithm we consider is a variant of the classical exponential mechanism (EM) known as posterior sampling. The posterior sampling algorithm instantiates the exponential mechanism by taking the quality score to be a scaled and regularized log-likelihood function with parameter $\gamma, \lambda$, i.e.,

$$\hat{\theta} \sim p(\theta) \propto \exp\left( -\gamma \left( \sum_{i=1}^n \ell_i(\theta) + \lambda\|\theta\|^2 \right) \right) \mathbf{1}(\theta \in \Theta). \tag{1}$$

This mechanism enjoys pure DP and Gaussian DP for appropriate choices of $\gamma, \lambda$, and domain $\Theta$.

**Lemma 4** (GDP of posterior sampling (Gopi et al., 2022, Theorem 4)). *Assume the loss function is $G$-Lipschitz, posterior sampling mechanism with parameter $\gamma, \lambda > 0$ satisfying $\gamma \leq \mu^2\lambda/G^2$ satisfies $\mu$-GDP.*

**Lemma 5** (Pure DP of posterior sampling). *Assume the loss function is $G$-Lipschitz, posterior sampling mechanism with parameter $\gamma > 0$ (any of $\lambda$) in a domain $\Theta$ satisfies $\varepsilon$-pure DP if $\gamma \leq \frac{\varepsilon}{G \cdot \mathrm{Diam}(\Theta)}$.*

Lemma 5 slightly improves existing analysis (Wang et al., 2015; Dimitrakakis et al., 2017) by the *bounded range* analysis (Dong et al., 2020). This approach avoids assumptions on the loss bound, which is potentially large, in privacy calculations. The proof is provided in Appendix I.4.

We have the following lemma for the utility of posterior sampling.

**Lemma 6** (De Klerk & Laurent (2018, Corollary 1)). *For a convex function $F(\theta)$, and a convex set $\Theta \subset \mathbb{R}^d$, $\hat{\theta} \sim p(\theta) \propto \exp(-\gamma F(\theta))$ satisfies that $\mathbb{E}[F(\hat{\theta})] \leq \min_{\theta \in \Theta} F(\theta) + \frac{d}{\gamma}$.*

## 3   TECHNICAL TOOLS

This section introduces tools that facilitate MCMC sampling for *pure* DP, including Lemma 8 that converts TV distance bound to $W_\infty$ bound, and Algorithm 1, an MCMC sampler that converges in $W_\infty$ distance.

### 3.1   OVERVIEW: WHY DO WE USE $W_\infty$ DISTANCE?

Suppose $p^*$ is a Gibbs posterior that satisfies pure DP. In practice, MCMC samplers generate an approximate sample $\tilde{\theta}$ from an approximate distribution $\tilde{p}$ with $d_{TV}(\tilde{p}, p^*) < \xi$, where $d_{\text{TV}}$ represents the TV distance. Notice that this TV distance guarantee does not grant pure DP for sampling from $\tilde{p}$. Instead, it provides only $(\varepsilon, \delta)$-DP with $\delta = (1 + e^\varepsilon)\xi > 0$, as elucidated in Proposition 3 of Wang et al. (2015).

To circumvent this challenge, ASAP (Algorithm 2) adopts a two-part approach to address the privacy of $\tilde{\theta}$. First, we decompose $\tilde{\theta}$ into two components:

$$\tilde{\theta} = \theta_* + (\tilde{\theta} - \theta_*), \text{ where } \theta_* \sim p^*.$$

Note that $\theta_* \sim p^*$ satisfies pure DP by existing works Wang et al. (2015). Our aim is to add noise to $(\tilde{\theta} - \theta_*)$ so the perturbed difference, $(\tilde{\theta} - \theta_*) + noise$, also maintains pure DP. This allows the leverage of the composition lemma (Lemma 14) to devise an MCMC sampling methodology that guarantees pure DP.

However, to release $(\tilde{\theta} - \theta_*)$ with pure DP by appending noise, it is crucial to determine an upper bound on the sensitivity of this quantity with a certainty of 1. This necessitates bounding the particle-wise distance $\|\tilde{\theta} - \theta_*\|_1$ (or $\|\tilde{\theta} - \theta_*\|_2$ in Gaussian DP) *with a probability of 1*. Establishing this sensitivity bound hinges on the Wasserstein-infinity distance (Definition 7), see Fiure 1 for a visual illustration of $W_\infty$.

Therefore, an MCMC sampling approach with $W_\infty$ error bounds is essential. We introduce MALA with constraint, specially designed to guarantee strong $W_\infty$ distance convergence.

### 3.2   TV DISTANCE TO $W_\infty$ DISTANCE

Consider two distributions $P$ and $Q$ on the same metric space $(\Theta, \text{dist})$. We sample $x \sim P$ and $y \sim Q$. When distributions $P$ and $Q$ are relatively "close", the corresponding samples $x$ and $y$ are also expected to be close. Motivated by the preceding discussion of MCMC for pure DP, a natural question arises: Can we define a distance for distributions $P$ and $Q$ that guarantees an upper bound for $\text{dist}(x, y)$ with probability 1? Wasserstein-$\infty$ distance answers this by constructing an appropriate coupling $\zeta^*$ between $P$ and $Q$.

**Definition 7.** *Let $\Theta \subseteq \mathbb{R}^d$ be a domain equipped with a metric $\text{dist}$. Let $P$ and $Q$ be two probability measures on $\Theta$. The Wasserstein-infinity distance between $P$ and $Q$ with respect to $\text{dist}$, denoted as $W_\infty(P, Q)$, is defined as*

$$W_\infty(P, Q) := \inf_{\zeta \in \Gamma(P,Q)} \operatorname*{ess\,sup}_{(x,y) \in (\Theta \times \Theta, \zeta)} \text{dist}(x, y) = \inf_{\zeta \in \Gamma(P,Q)} \{c \geq 0 | \zeta(\{(x, y) \in \Theta^2 | \text{dist}(x, y) \leq c\}) = 1\},$$

*where $\Gamma(P, Q)$ is the set of all couplings of $P$ and $Q$, i.e., the set of all joint probability distributions $\zeta$ with marginals $P$ and $Q$ respectively. The expression $\operatorname{ess\,sup}_{(x,y) \in (\Theta \times \Theta, \zeta)} \text{dist}(x, y)$ denotes the essential supremum of $\text{dist}(x, y)$ with respect to measure $\zeta$. It captures the supremum of $\text{dist}(x, y)$, excluding sets of $\zeta$-measure zero.*

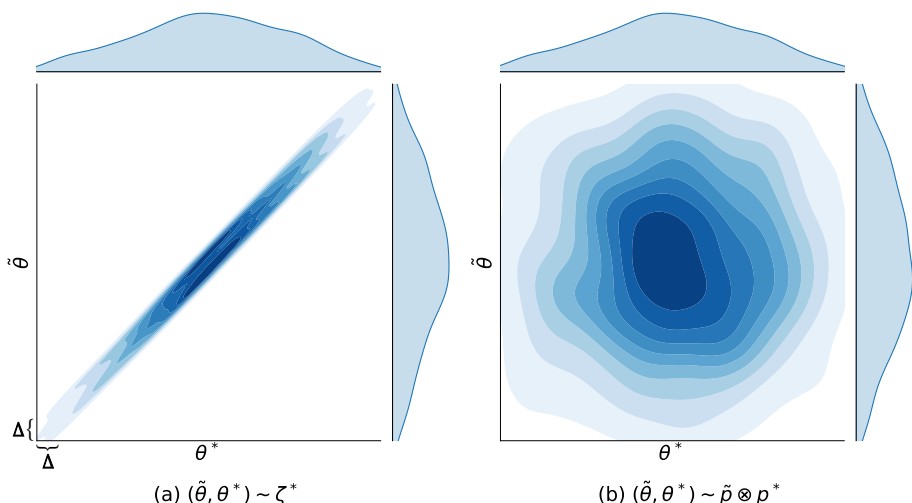

(a) $(\tilde{\theta}, \theta^*) \sim \zeta^*$        (b) $(\tilde{\theta}, \theta^*) \sim \tilde{p} \otimes p^*$

Figure 1: Two examples illustrating the couplings of $\tilde{p}$ and $p^*$. Let $\zeta^*$ be the optimal coupling of $W_\infty(\tilde{p}, p^*)$, and let $\tilde{p} \otimes p^*$ denote the independent coupling. In both scenarios, the marginal distributions are $\tilde{p}$ and $p^*$. Denote $\Delta = W_\infty(\tilde{p}, p^*)$. In Figure (a), when $(\tilde{\theta}, \theta^*)$ follows the optimal coupling, $(\tilde{\theta}, \theta^*)$ is confined within the band $|\tilde{\theta} - \theta^*| \leq \Delta$. Conversely, Figure (b) shows that when $\tilde{\theta}$ and $\theta^*$ are independently sampled, the distance $|\tilde{\theta} - \theta^*|$ can take relatively large values with positive probability. Through the appropriate coupling of the distributions $\tilde{p}$ and $p^*$, particularly via the optimal coupling $\zeta^*$, we obtain a tight almost-sure bound $\Delta$ on the distance between the two samples $\tilde{\theta}$ and $\theta^*$.

Our pursuit of pure DP relies on the $W_\infty$ distance. However, typical MCMC samplers offer convergence based on TV distance, which leads to the weaker approximate $(\varepsilon, \delta)$- DP with a $\delta > 0$. To reconcile this discrepancy, we present a versatile lemma that facilitates the conversion from $d_{\mathrm{TV}}$ bound to the $W_\infty$ bound.

**Lemma 8** (Converting TV distance to $W_\infty$ distance). *Consider two probability measures, $P$ and $Q$, both supported on a bounded closed 2-norm-ball $\Theta \subseteq \mathbb{R}^d$. On this domain $\Theta$, suppose that the density of $Q$ is lower-bounded by a constant $p_{\min}$. We establish the following results based on the choice of metric for $W_\infty$:*

*1. Suppose $W_\infty$ is defined with the 1-norm, i.e., $\mathrm{dist}(x,y) = \|x - y\|_1$.*

$$\text{If} \quad d_{TV}(P,Q) < p_{\min} \cdot \frac{\pi^{d/2}}{2^{d+1} \cdot \Gamma(\frac{d}{2} + 1) \cdot d^{d/2}} \Delta^d, \quad \text{then} \quad W_\infty(P,Q) \leq \Delta.$$

*2. Suppose $W_\infty$ is defined with the 2-norm, i.e., $\mathrm{dist}(x,y) = \|x - y\|_2$.*

$$\text{If} \quad d_{TV}(P,Q) < p_{\min} \cdot \frac{\pi^{d/2}}{2^{d+1} \cdot \Gamma(\frac{d}{2} + 1)} \Delta^d, \quad \text{then} \quad W_\infty(P,Q) \leq \Delta.$$

*In both cases, $\Gamma$ represents the gamma function.*

*The results naturally extend to the case where $\Theta$ is a bounded open 2-norm ball.*

The proof is detailed in Appendix E. This lemma bridges the gap between TV distance and $W_\infty$ distance, enabling the preservation of pure DP guarantees with TV distance-based MCMC samplers.

Conversion from TV distance bound to $W_\infty$ bound necessitates extra conditions on the probability measures. The following lemma establishes a lower bound (denoted as $p_{\min}$ in Lemma 8) for the density of distributions that are log-concave and log-Lipschitz continuous or smooth. The proof is provided in Appendix I.1.

**Lemma 9.** *Consider a distribution with density $p(\theta) = \frac{1}{Z}e^{-\gamma J(\theta)}$ supported on a bounded convex domain $\Theta$ with 2-norm diameter $R$, where $Z = \int_\Theta e^{-\gamma J(\theta)}\mathrm{d}\theta$ is the normalization constant.*

*Assuming that $J$ is convex and $\widetilde{G}$-Lipschitz continuous, we establish the following bound on the density:*

$$\inf_{\theta\in\Theta} p(\theta) \geq e^{-\gamma\widetilde{G}R} \cdot \frac{\Gamma(\frac{d}{2}+1)}{\pi^{d/2}(R/2)^d}.$$

*Alternatively, if $J$ is convex and $\beta$-Lipschitz smooth, the density bound holds for any $\tilde\theta \in \Theta$:*

$$\inf_{\theta\in\Theta} p(\theta) \geq e^{-\gamma\left(R\|\nabla J(\tilde\theta)\|+\beta R^2/2\right)} \cdot \frac{\Gamma(\frac{d}{2}+1)}{\pi^{d/2}(R/2)^d}.$$

## 3.3 MALA with Constraint

From the previous discussion, the approximate and target distribution should be close in the $W_\infty$ distance to maintain the pure DP. According to Lemma 8, this requirement translates into an *exponentially* small TV distance error. Therefore, we utilize MALA, an asymptotically unbiased sampler, which converges *exponentially* fast in TV distance accuracy. Lemma 8 also underscores the necessity of sampling within a bounded set, leading us to discard samples outside our predefined domain as described in Algorithm 1. To guarantee finite termination of the algorithm, we set a maximum restart number $\tau_{\max}$. We provide the expected runtime, which suggests that the algorithm typically concludes during the early trials.

**Notations.** Define the objective function $J(\theta) = \sum_{i=1}^n \ell_i(\theta)$, and its minimizer $\theta^* = \arg\min_{\theta\in\mathbb{R}^d} J(\theta)$. Let $\pi$ and $p^*$ be the unconstrained and constrained Gibbs distribution respectively with density $\pi(\theta) \propto e^{-\gamma J(\theta)}$, and $p^*(\theta) \propto e^{-\gamma J(\theta)}\mathbf{1}(\theta \in \Theta)$. Let $\kappa = \beta/\alpha$ be the condition number, where $\beta$ and $\alpha$ are defined in Assumption 1 and 2 respectively.

**Assumption 1** (Smoothness). Function $J$ is $n\beta$-Lipschitz smooth.

**Assumption 2** (Strong-Convexity). Function $J$ is $n\alpha$-strongly convex.

**Assumption 3** (Domain bound). The domain $\Theta$ is a convex set such that $\mathbb{B}(\theta^*, R_1) \subset \Theta$, for $R_1 \geq 8\sqrt{\frac{d}{\gamma n\alpha}}$.

**Theorem 1** (Mixing time in TV-distance). *Consider Algorithm 1 with initial distribution $p^0 \sim \mathcal{N}(0, \frac{1}{\gamma n\beta}\mathbb{I})$, step size $h^k = \Theta\left(\min\left\{\kappa^{-1/2}(\gamma n\beta)^{-1}(d\ln\kappa + \ln 1/\xi)^{-1/2}, \frac{1}{\gamma n\beta d}\right\}\right)$, number of iterations $K = \Theta\left((d\ln\kappa + \ln 1/\xi)\max\left\{\kappa^{3/2}\sqrt{d\ln\kappa + \ln 1/\xi}, d\kappa\right\}\right)$, and maximum number of restarts $\tau_{\max} = \Theta(\ln 1/\xi)$. Under Assumptions 1, 2, and 3, this algorithm converges to $\xi$-accuracy in $d_{TV}(p_{\mathrm{out}}, p^*)$, where $p_{\mathrm{out}}$ is the output distribution. The expected number of gradient queries to $J$ is given by:*

$$\Theta\left((d\ln\kappa + \ln 1/\xi)\max\left\{\kappa^{3/2}\sqrt{d\ln\kappa + \ln 1/\xi}, d\kappa\right\}\right).$$

**Corollary 10** (Mixing time in $W_\infty$ distance). *Consider the $W_\infty$ distance with respect to the 1-norm. Let the domain $\Theta$ be a ball of radius $B$. Under the same assumptions and conditions as Theorem 1 with choosing $\ln(1/\xi) = \Theta(\gamma n\beta B^2 + d\ln B + d\ln d + d\ln(1/\Delta))$, Algorithm 1 converges to $\Delta$-accuracy in $W_\infty(p_{\mathrm{out}}, p^*)$. The expected number of gradient queries to $J$ is given by:*

$$\Theta\left((d\ln(\kappa Bd) + d\ln(1/\Delta) + \gamma n\beta B^2)\max\left\{\kappa^{3/2}\sqrt{d\ln(\kappa Bd) + d\ln(1/\Delta) + \gamma n\beta B^2}, d\kappa\right\}\right).$$

*Each gradient query to the empirical risk $J$ translates to $n$ queries to the individual losses $\ell_i$'s.*

The proof of Theorem 1 and Corollary 10 is provided in Appendix F.

**Remark 11.** *According to the choice of parameters $B, \gamma$ in Table 2, we have $\gamma B^2 \sim \mathcal{O}(1/n)$. We note that the computation dependence on $d$ is $\widetilde{\Theta}(d^2)$, and the number of queries to the individual losses $\ell_i$'s is $\widetilde{\Theta}(n)$.*

In comparison, if we employ vanilla rejection sampling (Casella et al., 2004) with a proposal distribution of $\mathcal{N}(0, \frac{1}{\gamma n \beta}\mathbb{I})$, then the convergence time scales as $e^{\gamma n (\beta - \alpha) R_1^2} \geq e^{64(\kappa - 1)d}$, where $R_1$ is defined according to Assumption 3. This factor scales exponentially with the condition number $\kappa$ and the dimension $d$.

---

**Algorithm 1** Metropolis-adjusted Langevin algorithm (MALA) with constraint

---

1: Input: Stepsizes $\{h^k\}$, number of (inner-loop) iterations $K$, objective function $J$, domain $\Theta$, parameter $\gamma$, maximum number of restarts $\tau_{\max}$.
2: **for** $t = 0, 1, 2, \ldots, \tau_{\max} - 1$ **do**
3:     Sample $\theta^0$ according to distribution $p^0$
4:     **for** $k = 0, 1, 2, \ldots, K - 1$ **do**
5:         Sample $\theta^{k+1} \sim \mathcal{N}\left(\theta^k - h^k \gamma \nabla J(\theta^k), 2h^k \mathrm{I}\right)$
6:         Sample $u^{k+1} \sim \mathcal{U}[0, 1]$, denote $p(\cdot|\theta)$ as the density of $\mathcal{N}\left(\theta - h^k \gamma \nabla J(\theta), 2h^k \mathrm{I}\right)$
7:         **if** $\dfrac{p\left(\theta^k | \theta^{k+1}\right) \pi\left(\theta^k\right)}{p\left(\theta^{k+1} | \theta^k\right) \pi\left(\theta^{k+1}\right)} < u^{k+1}$ **then**
8:             $\theta^{k+1} \leftarrow \theta^k$
9:         **end if**
10:     **end for**
11:     **if** $\theta^K \in \Theta$ **then**                              ▷ Accept the sample
12:         Return $\theta^K$ and halt
13:     **end if**
14: **end for**
15: Return an arbitrary $\theta \in \Theta$

---

# 4   MAIN RESULTS: APPROXIMATE SAMPLE PERTURBATION (ASAP)

This section introduces Approximate Sample Perturbation, along with the end-to-end localized ASAP.

## 4.1   APPROXIMATE SAMPLE PERTURBATION (ASAP)

ASAP (Algorithm 2) is designed to maintain pure DP and pure Gaussian DP by smoothing out an MCMC sample. Theorem 2 establishes the pure DP and pure GDP guarantee for the ASAP algorithm.

Consider two randomized mechanisms: a reference mechanism $\mathcal{M}$ and another mechanism $\widetilde{\mathcal{M}}$. Assume that $\mathcal{M}$ satisfies pure DP or Gaussian DP, whereas $\widetilde{\mathcal{M}}$ does not inherently offer these DP guarantees. According to Theorem 2, if the distributions of $\mathcal{M}$ and $\widetilde{\mathcal{M}}$ have a bounded disparity in $W_\infty$ distance, then it is feasible to achieve pure DP (or Gaussian DP) by generating samples from $\widetilde{\mathcal{M}}$ and appending noise scaled in proportion to the $W_\infty$ bound.

---

**Algorithm 2** Approximate SAmple Perturbation (ASAP)

---

1: Input: Dataset $D$, reference randomized mechanism $\mathcal{M} : \mathcal{X}^* \to \mathcal{P}_\mathcal{U}$. $W_\infty$ error $\Delta$, a black box sampler $\widetilde{\mathcal{M}}$ such that $W_\infty(\widetilde{\mathcal{M}}(D), \mathcal{M}(D)) \leq \Delta$. Privacy parameter $\varepsilon'$ if pure DP ($\mu'$ if Gaussian DP).
2: Run the MCMC sampler: $\tilde{\theta} \sim \widetilde{\mathcal{M}}(D)$.
3: Return $\hat{\theta} = \tilde{\theta} + \mathcal{N}(0, \frac{4\Delta^2}{\mu'^2}\mathrm{I}_d)$ if GDP (or $\hat{\theta} = \tilde{\theta} + \mathbf{Z}$ with $Z_i \overset{i.i.d.}{\sim} \mathrm{Lap}(\frac{2\Delta}{\varepsilon'})$, $i = 1, ..., d$ if pure DP.)

---

**Theorem 2** (DP Guarantees for Approximate SAmple Perturbation). *Let $\mathcal{M}, \widetilde{\mathcal{M}}$ be randomized algorithms with output space $\Theta \subset \mathbb{R}^d$ that satisfy the following for any input dataset $D$:*

$$W_\infty(\mathcal{M}(D), \widetilde{\mathcal{M}}(D)) \le \Delta.$$

1. *Suppose we define $W_\infty$ with the 2-norm. If $\mathcal{M}$ satisfies $\mu$-GDP, then the procedure $\widetilde{\mathcal{M}}(D) + \mathcal{N}(0, \frac{4\Delta^2}{(\mu')^2}I_d)$ satisfies $\sqrt{\mu^2 + \mu'^2}$-GDP.*

2. *Suppose we define $W_\infty$ with the 1-norm. If $\mathcal{M}$ satisfies $\varepsilon$-DP, then the procedure $\widetilde{\mathcal{M}}(D) + \mathbf{Z}$ with $Z_i \sim \mathrm{Lap}(\frac{2\Delta}{\varepsilon'})$ i.i.d. for $i = 1, ..., d$, satisfies $(\varepsilon + \varepsilon')$-DP.*

The proof is provided in Appendix G. As demonstrated in Section 3.1 and the proof, a $W_\infty$ distance guarantee is essential to obtain pure DP and pure GDP with ASAP. Other weaker distance metrics such as TV distance and $W_2$ distance are generally insufficient for this purpose. Remarkably, as shown in Lemma 8, when the domain is a bounded ball and the density is bounded away from zero, we can convert a weak TV distance bound into a strong $W_\infty$ bound. This enables the application of Algorithm 2 across various scenarios.

### 4.2 Localized ASAP and the End-to-End Guarantees

Lemma 8 suggests that sampling should be conducted within a bounded 2-norm ball. Therefore, before sampling, we first localize to a bounded ball centered at the initial point $\theta_0$, denoted as $\Theta = \mathbb{B}(\theta_0, B) = \{\theta \mid \|\theta - \theta_0\|_2 \le B\}$. After this localization, we implement the ASAP algorithm within the defined bounded domain, referring to the approach as "Localized ASAP."

---

**Algorithm 3** End-to-End Localized ASAP

---

1: Input: Parameters $\gamma, \lambda = 0, B$ as given in Table 2. Wasserstein-infinity error $\Delta$. Dataset $D$, individual loss $\ell_i$'s satisfying $G$-Lipschitz continuity, $\beta$-smoothness, and $\alpha$-strong convexity. Privacy parameter $\mu'$ for GDP (or $\varepsilon'$ for pure DP) for ASAP. Denote $J(\theta) = \sum_{i=1}^n \ell_i(\theta) + \frac{\lambda}{2}\|\theta - \theta_0\|^2$.

2: Run the localization Algorithm 4, and assign its output to $\theta_0$.

3: Call ASAP (Algorithm 2) with inputs

$$\begin{cases} \Delta, D, \gamma, \text{ privacy parameter } \varepsilon' \text{ if pure DP } (\mu' \text{ if Gaussian DP}) \\ \widetilde{\mathcal{M}} \leftarrow \text{Algorithm 1 on the domain } \Theta = \{\theta \| \|\theta - \theta_0\|_2 \le B\}, \text{ setting } h_k, K, \tau_{\max} \text{ as per Table 3} \\ \mathcal{M} \leftarrow \text{Reference distribution with density } p_{\mathcal{M}(D)}(\theta) \propto e^{-\gamma J(\theta)} \mathbf{1}\{\|\theta - \theta_0\| \le B\}, \end{cases}$$

and assign the output to $\hat{\theta}$.

4: Return $\hat{\theta}$.

---

The computational efficiency of the sampler depends on the choices of $B$ and $\gamma$, as well as assumptions on the loss functions. The quality of the solution from the localized ASAP depends on the initialization $\theta_0$, and the associated choice of $B$. We instantiate these parameters concretely in Appendix B.

We provide the following privacy, utility, and computational guarantees for the end-to-end Algorithm 3.

**Theorem 3** (Guarantees for localized-ASAP). *Set $\theta_0, B, \gamma, \lambda$ as Table 2. Let $\hat{\theta}$ be the output of Algorithm 3. Set the pure DP (or Gaussian DP) parameters for the output perturbation and the ASAP to be $\varepsilon$ (or $\mu$). Then*

1. *$\hat{\theta}$ satisfies $3\varepsilon$-pure DP (or $\sqrt{3}\mu$-Gaussian DP).*

2. *In pure DP case, the empirical (total) risk satisfies*

$$\mathbb{E}[\mathcal{L}(\hat{\theta})] - \mathcal{L}(\theta^*) \le \mathcal{O}\left(\frac{d^2 G^2 (\ln d + \ln \kappa)}{\alpha n \varepsilon^2}\right).$$

> *In the Gaussian DP case, the empirical risk satisfies $\mathbb{E}[\mathcal{L}(\hat{\theta})] - \mathcal{L}(\theta^*) \leq \mathcal{O}\left(\frac{G^2(d+\ln\kappa)}{\alpha n\mu^2}\right)$.*

> 3. *The expected overall runtime time of the algorithm for pure DP is given by*
>
> $$\Theta\left(nd\left(\kappa\ln(d\kappa) + \ln n\right)\max\left\{\kappa^{3/2}\sqrt{d\left(\kappa\ln(d\kappa) + \ln n\right)}, d\kappa\right\}\ln(d\kappa)\right).$$
>
> *The expected overall runtime time of the algorithm for Gaussian DP is given by*
>
> $$\Theta\left(n\left(d\kappa + d\ln n + \kappa\ln\kappa\right)\max\left\{\kappa^{3/2}\sqrt{d\kappa + d\ln n + \kappa\ln\kappa}, d\kappa\right\}\ln(\kappa)\right).$$

The proof is presented in Appendix H.

**Remark 12.** *The suboptimality bound matches the information-theoretic limit for this problem in (Bassily et al., 2014) up to a constant factor. We enhance the $\kappa$ dependency in the empirical risk bound from $\mathcal{O}(\kappa)$ to $\mathcal{O}(\log\kappa)$. To our knowledge, this is the first $\tilde{\mathcal{O}}(n)$ time algorithm (assuming $\kappa = \text{polylog}(n)$) for DP-ERM (Lipschitz, smooth, strong convex losses) that achieves the optimal rates under pure DP. The nearest algorithm is from (Bassily et al., 2014), exhibiting a runtime of $\mathcal{O}(n^4)$. A nearly linear-time algorithm for DP-SCO in the smooth and strongly convex setting that achieves optimal rates exists (Feldman et al., 2020). But to the best of our knowledge, the problem remains open for DP-ERM until this paper.*

**Remark 13** (Improvement under Gaussian DP)**.** *The risk bound for pure DP, combined with the transition from pure DP to Gaussian DP (as per Lemma 15), inherently provides a DP-ERM learner under Gaussian DP with a suboptimal risk bound of $\tilde{\mathcal{O}}(\frac{d^2 G^2}{\alpha n\mu^2})$. However, we improve the dimensionality dependence in the empirical risk bound, from $\tilde{\mathcal{O}}\left(d^2\right)$ to $\mathcal{O}(d)$, under Gaussian DP with slightly less runtime.*

**Comparing to the existing work.** Current literature lacks a pure DP or Gaussian DP learner that attains the optimal rate for strongly convex problems in nearly linear time. The closest approach, Noisy Gradient Descent, runs in $\widetilde{\mathcal{O}}(n)$ time for strongly convex and smooth problems (with $\kappa\log n$ iterations), but its empirical risk is suboptimal by a factor of $\kappa\log n$. An alternative parameter regime that runs Noisy Gradient Descent for $\mathcal{O}(n^2)$ iterations (i.e., $\mathcal{O}(n^3)$ time) achieves the optimal rate (without additional $\kappa$ or $\log n$ dependence), albeit more slowly and without leveraging the smoothness. Further details are provided in Appendix J.

Other works either operate in a different setting or do not apply to all problems that we consider. For instance, objective perturbation with approximate minima perturbation (Iyengar et al., 2019) runs in $\widetilde{\mathcal{O}}(n)$ time but is restricted to generalized linear losses. Feldman et al. (2020) develop a nearly linear time DP learner using a "privacy amplification by iteration" technique, yet it works under the DP-SCO setting, which is different from DP-ERM that we consider. The same algorithm for DP-ERM still requires $\widetilde{\mathcal{O}}(n^2)$ time. In addition, their focus was on $(\varepsilon, \delta)$-DP. The sampler from (Gopi et al., 2022) (without combined with localization) operates in $\widetilde{\mathcal{O}}(\frac{n}{\alpha}\log(d/\delta))$ time, but becomes vacuous when aiming for $\delta = 0$ for pure DP or pure Gaussian DP.

## 5 CONCLUSION

Our proposed sampler Approximate SAample Perturbation (abbr. ASAP) perturbs an MCMC sample with noise proportional to its Wasserstein-$\infty$ distance from a reference distribution that satisfies *pure* DP or *pure* Gaussian DP. We show that our sampler obtains the first nearly linear-time end-to-end algorithm that achieves the optimal rate in the DP-ERM problem with strongly convex and smooth losses. The new techniques we developed might be of independent interest elsewhere that rely on approximate sampling.

**Limitations.** While the posterior sampling mechanism is known to achieve optimal rates in the general convex Lipschitz loss cases under pure DP and pure Gaussian DP (Gopi et al., 2022), a meticulous adaptation of our method is necessary to attain optimal rates in this scenario with fast computation. It remains an intriguing open problem to characterize the optimal computational complexity in settings that lack smoothness or strong convexity.

## ACKNOWLEDGEMENTS

The research is partially supported by the NSF awards: SCALE MoDL-2134209, CCF-2112665 (TILOS), as well as CNS-2048091 (CAREER). It is also supported by the U.S. Department of Energy, the Office of Science, the Facebook Research Award, a Google Research Scholar Award, as well as CDC-RFA-FT-23-0069 from the CDC's Center for Forecasting and Outbreak Analytics.

YL acknowledges Geelon So and Yuxing Huang for discussing the proofs in the paper. YW appreciates early conversations with Eric Vigoda, Zongchen Chen, and Daniel Štefankovič at a 2022 Summer School on MCMC in UCSB. Finally, the authors thank Jinshuo Dong for communicating Lemma 15 to us.

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

# Appendix

## Table of Contents

## A  ADDITIONAL PRELIMINARIES

**Lemma 14** (Adaptive Composition). *Let $\mathcal{M}_1 : \mathcal{X}^* \to \Delta^{\Theta_1}$ satisfies $\varepsilon_1$-DP (or $\mu_1$-GDP) and $\mathcal{M}_2 : \Theta_1 \times \mathcal{X}^* \to \Delta^{\Theta_2}$ : satisfies $\varepsilon_2$-DP (or $\mu_2$-GDP) with probability 1 when its first input $\theta_1 \sim \mathcal{M}_1(D)$ for any dataset $D \in \mathcal{X}^*$, then $\mathcal{M}_1 \times \mathcal{M}_2$ satisfies $(\varepsilon_1 + \varepsilon_2)$-DP (or $\sqrt{\mu_1^2 + \mu_2^2}$-GDP).*

The composition theorem traditionally requires $\mathcal{M}_2$ to satisfy DP (or GDP) for all $\theta_2 \in \Theta_2$. What is stated above is slightly weaker in the sense that we can ignore a measure 0 set. We defer the proof to Appendix I.3.

**Lemma 15** (Conversion between pure DP and GDP). *With $\Phi$ denoting the cumulative distribution function of the standard normal, an $\varepsilon$-pure DP mechanism also satisfies $\mu$-GDP with*

$$\mu = 2\Phi^{-1}\left(\frac{e^\varepsilon}{1 + e^\varepsilon}\right).$$

We defer the proof to Appendix I.5.

**Definition 16** (Exponential Mechanism McSherry & Talwar (2007)). *For a quality score $u$, the exponential mechanism $\mathcal{M}_u : \mathcal{X} \to \Delta^\Theta$ samples an outcome $\theta \in \Theta$ with probability proportional to $\exp\left(\frac{\varepsilon u(x,\theta)}{2\Delta u}\right)$, and is $\varepsilon$-DP, where $\Delta u$ is the global sensitivity.*

## B  LOCALIZATION TECHNIQUES

The choice for the algorithm for providing $\theta_0$ and the associated $B$ parameter in Algorithm 3 is delicate. We construct an appropriate algorithm to find $\theta_0$ in this section. In particular, we use approximate output perturbation.

For the general Lipschitz loss setting, output perturbation does not quite work as it does not even achieve minimax rate. But if individual loss function $\ell$ satisfies $\alpha$-strongly convexity, Bassily et al. (2014) showed that one can obtain a localization using output perturbation that qualifies such that the downstream exponential mechanism on the localized set is optimal. The key observation is that under strong convexity, the sensitivity of $\theta^* = \arg\min_\theta \sum_{i=1}^n \ell_i(\theta)$ can be shown to be $G/(\alpha n)$ and output perturbation satisfies pure-DP, also it satisfies (with high probability)

$$\|\theta_0 - \theta^*\| \leq \frac{2Gd\log(\cdot)}{\alpha n \varepsilon}.$$

This order suffices for the posterior sampling to achieve the optimal rate.

We provide the following Approximate Output Perturbation algorithm, wherein we introduce perturbations to the approximate optimizer.

---

**Algorithm 4** Approximate Output Perturbation

---

1: **Input:** individual losses $\{\ell_i\}_{i=1}^n$ satisfying $G$-Lipschitz continuity and $\beta$-smoothness; strong convexity parameter $\alpha$. Privacy parameter $\mu$ for GDP (or $\varepsilon$ for pure DP).
2: Denote $\mathcal{L}(\theta) := \sum_{i=1}^n \ell_i(\theta)$, and $\theta^* := \arg\min_\theta \mathcal{L}(\theta)$. Set $\tilde{\Delta} := \frac{2\tau}{n} + \frac{2G}{\alpha n}$.
3: Solve for $\theta_{\text{opt}}$ that satisfies $\|\theta_{\text{opt}} - \theta^*\|_2 \leq \frac{\tau}{n}$.      ▷ Instantiation: Gradient Descent or Newton Descent.
4: Output $\theta_0 = \theta_{\text{opt}} + \boldsymbol{Z}$, where $\boldsymbol{Z}_i \sim \text{Lap}(\frac{\sqrt{d}\tilde{\Delta}}{\varepsilon})$ for $\varepsilon$-pure DP ($\boldsymbol{Z}_i \sim \mathcal{N}(0, \frac{\tilde{\Delta}^2}{\mu^2})$ for $\mu$-GDP).

---

**Remark 17.** *For step 3, we make the optimization oracle a black-box algorithm that enjoys a linear convergence rate. Instantiation of the optimization oracle could be the Gradient Descent or the Newton Descent, which enjoy a linear convergence rate under the assumptions of strong convexity and Lipschitz*

*smoothness (or c-stable Hessian by the Theorem 2 of Karimireddy et al. (2018)). Note that $\|\tilde{\theta} - \theta^*\|_2 \leq \frac{\tau}{n}$ is not a stopping criteria of the optimization algorithm. Instead, it is a property that $\tilde{\theta}$ would fulfill under the stopping criteria the additional assumption on $\mathcal{L}$.*

The following lemma provides the bound for the sensitivity of the minimizer $\theta^*$, which helps bound the sensitivity of the approximate minimizer $\tilde{\theta}$.

**Lemma 18.** *Let $D \in \mathcal{X}^*$. Let $D' \in \mathcal{X}^*$ be a neighbor such that $\mathcal{L}_{D'} - \mathcal{L}_D = \pm \ell_x$ for some $x \in \mathcal{X}$, then*

$$\|\theta^*(D) - \theta^*(D')\|_2 \leq \frac{2G}{\alpha n}.$$

With Lemma 18, we can bound the sensitivity of $\tilde{\theta}$, thus providing the following differential privacy guarantees.

**Lemma 19.** *Algorithm 4 satisfies $\varepsilon$-differential privacy (or $\mu$-Gaussian DP).*

We defer the proofs of Lemma 18 and 19 to Appendix I.2.

## C PARAMETERS IN LOCALIZED-ASAP

**The choice of $\gamma$ and $\lambda$.** Larger $\gamma$ provides a better utility guarantee but a weaker privacy guarantee. We choose $\gamma$ by the following fact so that the posterior sampling mechanism $\theta \sim p^*(\theta)$ satisfies $\varepsilon$-pure DP or $\mu$-GDP.

**Fact 1** (Optimal rates for strongly convex problems). *The optimal (expected) excess empirical risks for $G$-Lipschitz continuous and $\alpha$-strongly convex (individual) losses is $\frac{d^2 G^2}{\alpha n \varepsilon^2}$ and $\frac{d G^2}{\alpha n \mu^2}$ for $\varepsilon$-DP and $\mu$-GDP respectively. For posterior sampling, choosing*

$$\gamma = \frac{\mu^2 \alpha n}{G^2}, \quad \lambda = 0$$

*yields the optimal rate under $\mu$-GDP. Unfortunately, there are no parameter choices for $\gamma, \lambda$ that can yield the optimal rate under pure-DP, unless $\Theta$ is already localized such that it satisfies that $\mathrm{Diam}(\Theta) \leq \frac{dG}{\alpha n \varepsilon}$. Then the choice of $\gamma = \frac{\varepsilon}{G \cdot \mathrm{Diam}(\Theta)}$ works.*

**The choice of $B$ (the radius of the localized domain).** For desirable the utility and computational guarantees in sampling, we hope that $B$ is large enough such that the mode $\theta^*$ and the mean $\bar{\theta}$ are in the localized domain $\Theta$, i.e., $\theta^* \in \Theta$ and $\bar{\theta} \in \Theta$, where $\theta^* := \arg\min_{\theta \in \mathbb{R}^d} J(\theta)$, and $\bar{\theta} := \mathbb{E}_{\theta \sim \pi(\theta) \propto e^{-\gamma J(\theta)}} \theta$.

Moreover, Assumption 3 requires $\mathbb{B}(\theta^*, R_1) \subset \Theta$ for efficient constrained MALA, where $R_1 \geq 8\sqrt{\frac{d}{\gamma \alpha n}}$. However, by Corollary 20 and the discussion after Theorem 1, we know that an excessively large $B$ should be avoided. This is because a large $B$ leads to a reduced $p_{\min}$, which in turn necessitates a greater number of steps for constrained MALA to achieve the desired accuracy. Also, note that by Fact 1, for $\varepsilon$-pure DP guarantee, we set $\gamma = \frac{\varepsilon}{2GB}$, which is dependent of $B$.

In pure DP case, suppose $\theta_0$ is the output of Alogrithm 4, then we have $\theta_0 = \theta_{\mathrm{opt}} + \mathbf{Z}$, where $Z_i \sim \mathrm{Lap}(2\sqrt{d}\left(\tau + \frac{G}{\alpha}\right)/n\varepsilon)$, and $\|\theta_{\mathrm{opt}} - \theta^*\|_2 \leq \frac{\tau}{n}$. Therefore we have

$$\|\theta_0 - \theta^*\| \leq \|\theta_0 - \theta_{\mathrm{opt}}\|_2 + \|\theta_{\mathrm{opt}} - \theta^*\|_2 = \|\mathbf{Z}\|_2 + \|\tilde{\theta} - \theta^*\|_2 \leq \frac{2d\left(G + \tau\alpha\right)\ln(d/\rho)}{n\alpha\varepsilon} + \frac{\tau}{n},$$

with probability $1 - \rho$. In the last inequality, we apply the fact that for $Y_i \overset{\mathrm{iid}}{\sim} \mathrm{Lap}(\lambda)$, we have

$$\mathbb{P}\left(\|\mathbf{Y}\|_2 \geq \sqrt{d}\lambda\ln(d/\rho)\right) = \mathbb{P}\left(\sum_{i=1}^d Y_i^2 \geq d(\lambda\ln(d/\rho))^2\right) \leq \sum_{i=1}^d \mathbb{P}\left(|Y_i| \geq \lambda\ln(d/\rho)\right) \leq \rho.$$

In order to fulfill the condition $\mathbb{B}(\theta^*, R_1) \subset \Theta$, where $R_1 = 8\sqrt{\frac{d}{\gamma\alpha n}}$, note that $\gamma = \frac{\varepsilon}{2GB}$, the following inequality should hold:

$$B - 8\sqrt{\frac{2dGB}{\varepsilon\alpha n}} \geq \frac{2d(G + \tau\alpha)\ln(d/\rho)}{n\alpha\varepsilon} + \frac{\tau}{n}. \tag{2}$$

The above inequality is satisfied with $B \geq \frac{8(32dG + (\alpha\tau + G)d\ln(d/\rho) + \tau\varepsilon\alpha)}{\alpha n\varepsilon}$. For the sake of simplicity, we let $\tau$ be sufficiently small and set $B = \frac{C_1 dG\ln(d/\rho)}{\alpha n\varepsilon}$.

In GDP case, similarly, we have $\theta_0 = \theta_{\mathrm{opt}} + \boldsymbol{Z}$, where $Z_i \sim \mathcal{N}(0, 4(\tau + \frac{G}{\alpha})^2/n^2\mu^2)$, and $\|\theta_{\mathrm{opt}} - \theta^*\|_2 \leq \frac{\tau}{n}$. Therefore we have

$$\|\theta_0 - \theta^*\| \leq \|\theta_0 - \theta_{\mathrm{opt}}\|_2 + \|\theta_{\mathrm{opt}} - \theta^*\|_2 \leq \|\boldsymbol{Z}\|_2 + \frac{\tau}{n} \leq \frac{2\sqrt{2}(\tau\alpha + G)(\sqrt{d} + \sqrt{\ln(1/\rho)})}{\alpha n\mu} + \frac{\tau}{n}, \tag{3}$$

with probability $1 - \rho$. In the last inequality where we use the tail bound for chi-square distributions by Lemma 1 in (Laurent & Massart, 2000). Again we need $B - R_1 \geq \|\theta_0 - \theta^*\|$, where $R_1 = 8\sqrt{\frac{d}{\gamma\alpha n}}$ and $\gamma = \frac{\mu^2\alpha n}{G^2}$. This is satisfied with $B \geq \frac{2\sqrt{2}(\tau\alpha + G)(\sqrt{d} + \sqrt{\ln(1/\rho)}) + 8G\sqrt{d} + \tau\alpha\mu}{\alpha n\mu}$. Again, we let $\tau$ be sufficiently small and set $B = \frac{C_2 G(\sqrt{d} + \sqrt{\ln(1/\rho)})}{\alpha n\mu}$

**Calculation of $p_{\min}$**   Lemma 9 implies that the choices of $p_{\min}$ in Algorithm 3 are valid lower bounds of the density function. We have the following corollary.

**Corollary 20.** *Consider probability with density*

$$p(\theta) \propto \exp\left(-\gamma\left(\sum_{i=1}^{n} \ell_i(\theta) + \frac{\lambda}{2}\|\theta - \theta_0\|^2\right)\right)\mathbf{1}\{\|\theta - \theta_0\| \leq B\}.$$

1. *If $\ell_i$ is G-Lipschitz for all $i$, then $\min_\theta p(\theta) \geq e^{-\gamma(2nGB + 2\lambda B^2)}\frac{\Gamma(\frac{d}{2}+1)}{\pi^{d/2}B^d}$.*

2. *If $\ell_i$ is convex and $\beta$-smooth for all $i$, then for all $\tilde{\theta} \in \Theta$, we have*

$$\min_\theta p(\theta) \geq e^{-\gamma\left(2B\|\nabla J(\tilde{\theta})\| + 2(n\beta + \lambda)B^2\right)} \cdot \frac{\Gamma(\frac{d}{2} + 1)}{\pi^{d/2}B^d}.$$

Applying Corollary 20, we calculate $p_{\min}$ as follows: Take any $\theta'$ such that $\|\theta' - \theta_0\|_2 \leq B$. Compute

$$p_{\min} \leftarrow e^{-\gamma\min\left\{2nGB + 2\lambda B^2, \, 2\|\nabla L_\lambda(\theta')\|B + 2(n\beta + \lambda)B^2\right\}}\frac{\Gamma(\frac{d}{2}+1)}{\pi^{d/2}B^d}.$$

# D  EXPERIMENTS

## D.1  THEORETICAL LOWER BOUNDS

We visualize the excess empirical risks in Figure 2and demonstrate that $\varepsilon$-(pure) DP can outperform $(\varepsilon, \delta)$-DP in theory, especially when the sample size $n$ is large.

## D.2  EMPIRICAL RISKS ON REAL DATASETS

**Setup.** We experiment on two real datasets: Red Wine Quality and White Wine Quality from UCI repository (https://archive.ics.uci.edu/dataset/186/wine+quality). The Red Wine Quality dataset contains 1599 samples with 11 features. We standardize the data and learn a regression task via $\ell_i(\theta) = \frac{1}{2}(x_i^T\theta - y_i)^2 + \frac{\alpha}{2}\|\theta\|^2$, which is a strongly convex problem under $\alpha = 100$ and $\rho = 0.01$. The

Table 1: Summary of our results for the $\alpha n$-strong convex DP-ERM problem with $G$-Lipschitz and $\beta$-smooth losses and that the total loss satisfies $\alpha n$-strong convexity. The utility measures the expected excess empirical (total) risk. The $\mathrm{polylog}(n)$ factors are ignored from the computation. Observe that we are the first algorithms that achieve the nearly linear time computation under pure-DP and pure-Gaussian DP while still obtaining the optimal excess empirical risk.

| | Privacy | Excess Empirical risk | Computation |
|---|---|---|---|
| NoisyGD (see Appendix J) | $\mu$-GDP | $dG^2/\alpha n\mu^2$ | $n^3$ |
| NoisySGD (Bassily et al., 2014) | $(\varepsilon, \delta)$-DP | $dG^2 \log(1/\delta)/\alpha n\varepsilon^2$ | $n^2$ |
| PosteriorSample (Gopi et al., 2022) | $\delta$-approx $\mu$-GDP | $dG^2/\alpha n\mu^2$ | $n/\alpha$ |
| ExponentialMechanism (Bassily et al., 2014) | $\varepsilon$-DP | $d^2G^2/\alpha n\varepsilon^2$ | $n^4$ |
| Localized-PS-ASAP (This paper) | $\mu$-GDP | $dG^2/\alpha n\mu^2$ | $n/\alpha^2$ or $n/\alpha^{5/2}$ |
| Localized-EM-ASAP (This paper) | $\varepsilon$-DP | $d^2G^2/\alpha n\varepsilon^2$ | $n/\alpha^2$ or $n/\alpha^{5/2}$ |

Table 2: The choice of $\gamma, B, \lambda, \theta_0, \Delta, \rho$ when instantiating Algorithm 3 for pure DP or Gaussian DP learning. The choices of $\gamma$ ensure the DP and GDP guarantee respectively and the choice of $B$ ensures sufficient localization such that the sampling Algorithm 1 run in $\tilde{O}(n)$ time (condition on the high-prob event that $\theta^*$ is inside the localized set $\Theta$. Polynomial dependence in $d$ and other parameters are hidden in $\tilde{O}$). For pure-DP, the choice of $B$ has the additional purpose of achieving the optimal rate for DP-ERM — $\mathcal{L}(\hat{\theta}) - \mathcal{L}(\theta^*) = O(\frac{d^2G^2}{\alpha n\varepsilon^2})$. For GDP, the choice of $B$ only affects computation. The optimal rate $\mathcal{L}(\hat{\theta}) - \mathcal{L}(\theta^*) = O(\frac{dG^2}{\alpha n\mu^2})$ is always attained.

| | $\gamma$ | $B$ | $\lambda$ | $\theta_0$ | $\Delta$ | $\rho$ |
|---|---|---|---|---|---|---|
| $\varepsilon$-Pure DP | $\frac{\varepsilon}{2GB}$ | $\frac{C_1 G d \ln(d/\rho)}{\alpha n\varepsilon}$ | $0$ | Algorithm 4 | $\frac{dG\ln(d/\rho)}{2n^2\alpha\varepsilon}$ | $\frac{1}{d\kappa}$ |
| $\mu$-GDP | $\frac{\mu^2\alpha n}{G^2}$ | $\frac{C_2 G(\sqrt{d}+\sqrt{\ln(1/\rho)})}{\alpha n\mu}$ | $0$ | Algorithm 4 | $\frac{\sqrt{d}G}{\sqrt{2}n^2\alpha\mu}$ | $\frac{1}{d\kappa}$ |

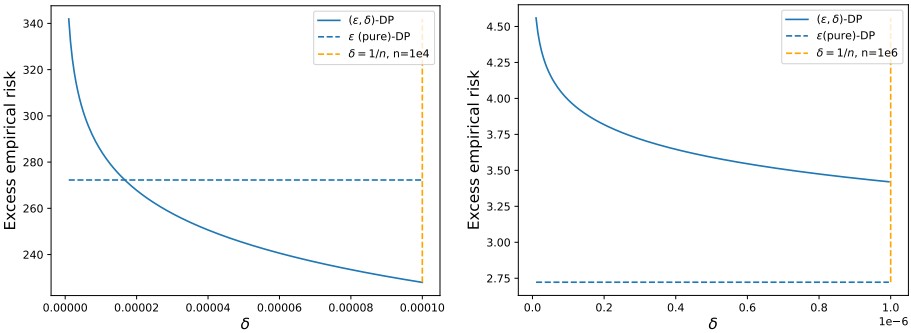

Figure 2: Excess empirical risks from Table 1 for strongly convex losses. Here $d = 11, G = 300, \alpha = 4, \varepsilon = 1$. Left $n = 1e4$. Right $n = 1e6$.

Table 3: Choices of step sizes, number of iterations, and maximum number of restarts in Algorithm 1 for pure DP and Gaussian DP.

| Pure Differential Privacy | |
|---|---|
| $h_k$ | $\Theta\left( \frac{G^2 \ln(d/\rho)}{\alpha \beta n^2 \varepsilon^2} \min\left\{ \kappa^{-\frac{1}{2}} d^{\frac{1}{2}} \left( \kappa \ln(d\kappa) + \ln n \right)^{-\frac{1}{2}}, 1 \right\} \right)$ |
| $K$ | $\Theta\left( d \left( \kappa \ln(d\kappa) + \ln n \right) \max\left\{ \kappa^{\frac{3}{2}} \sqrt{d \left( \kappa \ln(d\kappa) + \ln n \right)}, d\kappa \right\} \right)$ |
| $\tau_{\max}$ | $\Theta\left( d \left( \kappa \ln(d\kappa) + \ln n \right) \right)$ |
| Gaussian Differential Privacy | |
| $h_k$ | $\Theta\left( \frac{G^2}{\alpha \beta n^2 \mu^2} \min\left\{ \kappa^{-\frac{1}{2}} \left( d\kappa + d \ln n + \kappa \ln \kappa \right)^{-\frac{1}{2}}, \frac{1}{d} \right\} \right)$ |
| $K$ | $\Theta\left( (d\kappa + d \ln n + \kappa \ln \kappa) \max\left\{ \kappa^{3/2} \sqrt{d\kappa + d \ln n + \kappa \ln \kappa}, d\kappa \right\} \right)$ |
| $\tau_{\max}$ | $\Theta\left( d\kappa + d \ln n + \kappa \ln \kappa \right)$ |

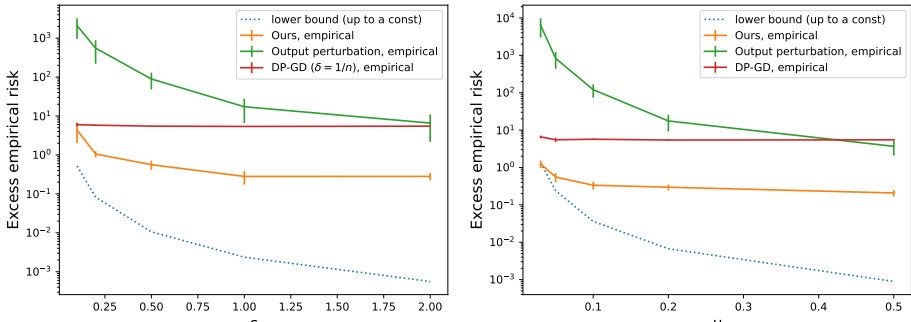

Figure 3: Excess empirical risks for strongly convex losses on Wine Quality – Red dataset.

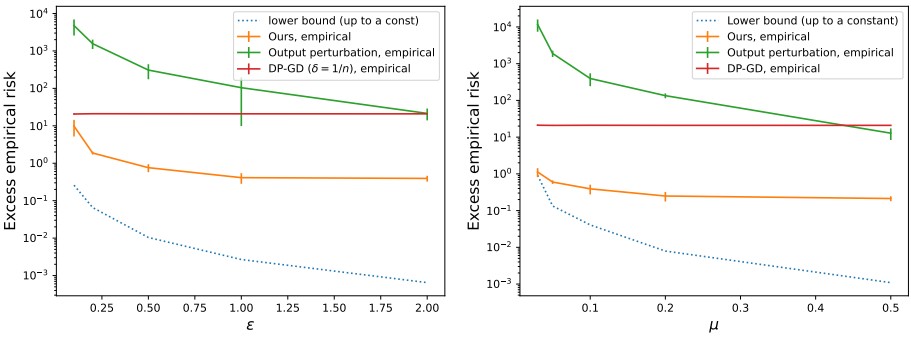

Figure 4: Excess empirical risks for strongly convex losses on Wine Quality – White dataset.

White Wine Quality dataset contains 4898 samples with 11 features. We standardize the data and learn a regression task via $\ell_i = \frac{1}{2}(x_i^T \theta - y_i)^2 + \frac{\alpha}{2}\|\theta\|^2$, which is a strongly convex problem under $\alpha = 32$ and $\rho = 0.01$.

**Results.** We compare with the theoretical lower bounds in Table 1 (up to a constant), the output perturbation in Algorithm 4, and DP-GD with per-sample gradient clipping (no need to set the clipping threshold) by Bu et al. (2022). The experiments are conducted under both $\varepsilon$-DP and $\mu$-GDP. We consistently observe that our algorithms outperform the existing ones.

## E PROOF THE CONVERSION LEMMA 8

### E.1 A KEY LEMMA: ANOTHER CHARACTERIZATION OF $W_\infty$ DISTANCE

To establish Lemma 8, we commence by furnishing an equivalent characterization for the $W_\infty$ distance. We also demonstrate the attainability of the infimum in Lemma 22. Our exposition begins with the introduction of the concept of tightness.

**Definition 21** (Tightness). *A probability measure $P$ on $\Theta$ is called tight if for any $\epsilon > 0$, there exists a compact set $K \subset \Theta$, such that*

$$P(\Theta \setminus K) \leq \epsilon,$$

**Lemma 22** (Proposition 5 and Proposition 1 of Givens & Shortt (1984)). *Let $(\Theta, \mathrm{dist})$ be a complete separable metric space. Let $P$ and $Q$ be two tight probability measures on $\Theta$. The Wasserstein-$\infty$ distance has the following characterization:*

$$W_\infty(P, Q) = \inf\{r > 0 : P(U) \leq Q(U^r), \text{ for all open subsets } U \subset \Theta\},$$

*where the $r$-expansion of $U$ is denoted by $U^r := \{x \in \Theta : \mathrm{dist}(x, U) \leq r\}$.*

*Furthermore, the infimum in the Definition 7 can be attained, i.e., there exists $\zeta^* \in \Gamma(P, Q)$ such that*

$$W_\infty(P, Q) = \operatorname*{ess\,sup}_{(x,y) \in (\Theta \times \Theta, \zeta^*)} \mathrm{dist}(x, y) = \inf_{\zeta \in \Gamma(P,Q)} \operatorname*{ess\,sup}_{(x,y) \in (\Theta \times \Theta, \zeta)} \mathrm{dist}(x, y).$$

The proof of this lemma is presented in Givens & Shortt (1984). For completeness, we provide a detailed proof in Appendix I.6.

### E.2 PROOF OF LEMMA 8

**Proof of Lemma 8** First note that $\|x - y\|_1 \leq \sqrt{d}\|x - y\|_2$, for all $x, y$. Therefore, it suffices to prove the lemma when $\mathrm{dist}$ is the $\ell_2$ metric.

Fix $\Delta$ and set $\xi = p_{\min} \cdot \frac{\pi^{d/2}}{2^{d+1} \cdot \Gamma(d/2+1)}\Delta^d$, so that $d_{\mathrm{TV}}(P, Q) < \xi$.

To prove the result that $W_\infty \leq \Delta$, we use the equivalent definition of $W_\infty$ in Lemma 22. By this definition, to prove $W_\infty(P, Q) \leq \Delta$, it suffices to show that

$$P(A) \leq Q(A^\Delta), \text{ for all open set } A \subseteq \Theta,$$

where we re-define $A^\Delta := \{x \in \mathbb{R}^d \mid \mathrm{dist}(x, A) \leq \Delta\}$. Note that by this definition, $A^\Delta$ might extend beyond $\Theta$. However, we still have $Q(A^\Delta) = Q(A^\Delta \cap \Theta)$, since $Q$ is supported on $\Theta$.

Note that if $Q(A^\Delta) = 1$, it is obvious that $P(A) \leq Q(A^\Delta)$. When $A$ is an empty set, the proof is trivial. So we only consider nonempty open set $A \subseteq \Theta$ with $Q(A^\Delta) < 1$.

Note that for an arbitrary open set $A \subseteq \Theta$, we have

$$P(A) \leq Q(A) + d_{TV}(P, Q) < Q(A) + \xi.$$

Thus to prove $P(A) \leq Q(A^\Delta)$, it suffices to prove $Q(A) + \xi \leq Q(A^\Delta)$, i.e., $Q(A^\Delta \setminus A) \geq \xi$.

To prove $Q(A^\Delta \setminus A) \geq \xi$, we construct an $\ell_2$-ball $U \subset \mathbb{R}^d$ of radius $\Delta/2$ satisfying the properties:

1. $U$ is contained in the set $A^\Delta \setminus A$, i.e., $U \subset A^\Delta \setminus A$, and

2. $Q(U) = Q(U \cap \Theta) \geq \xi$

Notice that we only consider nonempty open set $A \subseteq \Theta$ with $Q(A^\Delta) < 1$, we construct $U$ as follows.

**Intuition and Overview of the construction of $U$.** The construction of the ball $U$ involves determining two pivotal points, $y_\Delta$ and $y_A$, located at the "boundaries" of $A^\Delta$ and $A$, respectively. We then define $U$ as $U := \mathbb{B}_{\text{dist}}\left(\frac{y_\Delta + y_A}{2}, \frac{\text{dist}(y_\Delta, y_A)}{2}\right)$. Note that by the definition of $A^\Delta$, $\text{dist}(y_\Delta, y_A) \geq \Delta$. To guarantee Property 1 (i.e., $U \subset A^\Delta \setminus A$), $y_A$ is chosen such that $\text{dist}(y_\Delta, y_A) = \Delta$. Additionally, to ensure Property 2 (i.e., $Q(U) \geq \xi$), since $Q(U) = Q(U \cap \Theta)$, our aim is to maximize the "size" of $U \cap \Theta$. Therefore, we specifically require $y_\Delta$ to belong to $\Theta$, while noting that $A^\Delta$ might extend beyond $\Theta$. The detailed selection process of $y_\Delta$ and $y_A$, and the subsequent construction of $U$, are presented below.

**Existence and Construction of $y_\Delta$.** To find the above discussed $y_\Delta \in \Theta$, we first show that:

$$\text{If } Q(A^\Delta) < 1, \text{ then } \partial(A^\Delta) \cap \Theta \neq \emptyset,$$

where we re-define the boundary of $A^\Delta$ as $\partial(A^\Delta) := \{x \in \mathbb{R} \mid \text{dist}(x, A) = \Delta\}$.

We prove it by contradiction. If instead $\partial(A^\Delta) \cap \Theta = \emptyset$, then for all $x \in \Theta$, $\text{dist}(x, A) \neq \Delta$. Due to the continuity of dist and the convexity of $\Theta$, we know that only one of these two statements holds:

- $\text{dist}(x, A) < \Delta$, for all $x \in \Theta$.

- $\text{dist}(x, A) > \Delta$, for all $x \in \Theta$.

Since $\emptyset \neq A \subseteq \Theta$, there exist $x' \in A \in \Theta$, such that $\text{dist}(x', A) = 0$. Therefore, the first statement holds. Thus $\Theta \subseteq A^\Delta$, which contradicts to $Q(A^\Delta) < 1$.

Therefore, $\partial(A^\Delta) \cap \Theta \neq \emptyset$. Then there exists $y_\Delta \in \partial(A^\Delta) \cap \Theta$. Thus there exist $y_\Delta \in \Theta$, such that $\text{dist}(y_\Delta, A) = \Delta$.

**Construction of $y_A$.** Note that by the definition of $\text{dist}(\cdot, A)$ and that $\text{dist}(y_\Delta, A) = \Delta$, there exist $y_A \in \overline{A}$ (the closure of $A$), such that $\text{dist}(y_\Delta, y_A) = \Delta$.

**Construction of $U$.** Let $y_0 = \frac{y_\Delta + y_A}{2}$ and let $U$ be the closed ball $U = \mathbb{B}(y_0, \Delta/2) := \{x \in \mathbb{R}^d \mid \text{dist}(x, y_0) \leq \Delta/2\}$. We now prove that $U \subseteq A^\Delta \setminus A$ and that $Q(U) \geq \xi$, and then the proof is concluded.

1. To prove $U \subset A^\Delta \setminus A$, let $x \in U$. We show that $x \in A^\Delta$ and $x \notin A$. Since $x, y_\Delta, y_A \in U$ and $U$ is a ball with diameter $\Delta$, we have

$$\text{dist}(x, y_\Delta) \leq \Delta, \text{ and } \text{dist}(x, y_A) \leq \Delta \tag{4}$$

- $(x \notin A)$. If $x \in A$, since $A$ is an open set and $\text{dist}(y_\Delta, A) = \Delta$, we have that $\text{dist}(x, y_\Delta) > \Delta$, which contradicts to (4). Therefore $x \notin A$.

- $(x \in A^\Delta)$. Note that $y_A \in \overline{A}$, thus $\text{dist}(x, A) \leq \text{dist}(x, y_A) \overset{(4)}{\leq} \Delta$, implying that $x \in A^\Delta$.

2. To prove $Q(U) \geq \xi$, note that we assume the domain $\Theta$ to be an $\ell_2$ ball. Since $y_A, y_\Delta \in \Theta$, by Lemma 24, we know that $\text{Vol}(U \cap \Theta) \geq \frac{1}{2}\text{Vol}(U)$. Thus

$$Q(U) = Q(U \cap \Theta) \geq p_{\min}\text{Vol}(U \cap \Theta) \geq \frac{1}{2}p_{\min}\text{Vol}(U) = \xi,$$

which completes the proof.

**Remark 23.** *It is worth noting that Lemma 8 and Lemma 24 extend naturally to open balls. The necessary modifications in the proof involve changing certain inequalities: "≤" becomes "<", and ">" becomes "≥."*

∎

**Lemma 24.** *Suppose $\theta \in \mathbb{R}^d$, and $R > 0$. If there exists $a \in \mathbb{R}^d$ such that $a \in \mathbb{B}_{\ell_2}(\theta, R)$, and that $-a \in \mathbb{B}_{\ell_2}(\theta, R)$. Then*

$$\frac{\text{Vol}\left(\mathbb{B}_{\ell_2}(0, \|a\|_2) \cap \mathbb{B}_{\ell_2}(\theta, R)\right)}{\text{Vol}\left(\mathbb{B}_{\ell_2}(0, \|a\|_2)\right)} \geq \frac{1}{2}.$$

**Proof** Since $a, -a \in \mathbb{B}_{\ell_2}(\theta, R)$, we have

$$\|a - \theta\|_2^2 \leq R^2, \text{ and } \| - a - \theta\|_2^2 \leq R^2,$$

which implies

$$\|a\|_2^2 - 2\langle a, \theta \rangle + \|\theta\|_2^2 \leq R^2, \text{ and } \|a\|_2^2 + 2\langle a, \theta \rangle + \|\theta\|_2^2 \leq R^2.$$

Thus, $\|a\|_2^2 + \|\theta\|_2^2 \leq R^2$.

Denote $S = \mathbb{B}_{\ell_2}(0, \|a\|_2) \setminus \mathbb{B}_{\ell_2}(\theta, R)$. We first prove that $-S := \{-x \mid x \in S\} \subseteq \mathbb{B}_{\ell_2}(\theta, R)$. For an arbitrary $x \in S$, since $x \in \mathbb{B}_{\ell_2}(0, \|a\|_2)$, we have $\|x\|_2 \leq \|a\|_2$, thus $\|x\|_2^2 + \|\theta\|_2^2 \leq \|a\|_2^2 + \|\theta\|_2^2 \leq R^2$. On the other hand, since $x \notin \mathbb{B}_{\ell_2}(\theta, R)$, we have $\|x - \theta\|_2^2 > R^2$. Thus $\| - x - \theta\|_2^2 = 2\left(\|x\|_2^2 + \|\theta\|_2^2\right) - \|x - \theta\|_2^2 < R^2$, which implies $-x \in \mathbb{B}_{\ell_2}(\theta, R)$. Therefore $-S \subseteq \mathbb{B}_{\ell_2}(\theta, R)$.

Since $S = \mathbb{B}_{\ell_2}(0, \|a\|_2) \setminus \mathbb{B}_{\ell_2}(\theta, R)$ and $-S \subseteq \mathbb{B}_{\ell_2}(\theta, R)$, we know that $S \cap (-S) = \emptyset$. Therefore,

$$2\text{Vol}(S) = \text{Vol}(S) + \text{Vol}(-S) = \text{Vol}\left(S \cup (-S)\right) \leq \text{Vol}\left(\mathbb{B}_{\ell_2}(0, \|a\|_2)\right).$$

Therefore

$$\text{Vol}\left(\mathbb{B}_{\ell_2}(0, \|a\|_2) \cap \mathbb{B}_{\ell_2}(\theta, R)\right) = \text{Vol}\left(\mathbb{B}_{\ell_2}(0, \|a\|_2)\right) - \text{Vol}(S) \geq \frac{1}{2}\text{Vol}\left(\mathbb{B}_{\ell_2}(0, \|a\|_2)\right).$$

∎

## F    PROOFS OF THEOREM 1 AND COROLLARY 10

In this section, we provide the omitted proofs and supplementary facts for the Metropolis-adjusted Langevin algorithm (MALA) with constraint in Section 3.3.

We first provide the definition of mixing time. Given an error tolerance $\xi \in (0, 1)$ and an initial distribution $p_0$, define the $\xi$-mixing time in total variation distance as

$$\tau\left(\xi, p^0, p^*\right) = \min\{k \mid \|p^k - p^*\|_{\text{TV}} \leq \xi\}$$

For simplicity, we denote $L = n\beta$ and $m = n\alpha$ in this proof.

**Proof of Theorem 1** We separate the proof of Theorem 1 into three parts via splitting of Algorithm 1 into the inner loop, the outer loop, and the case where the outer loop count reaches the maximum. We note that in the inner loop, Algorithm 1 is performing the MALA algorithm on the unconstrained space $\mathbb{R}^d$. In the outer loop, it performs a rejection sampling step that takes in independent samples from MALA and keeps rejecting samples outside of the domain $\Theta$ until one sample falls inside $\Theta$. When the sample keeps falling outside $\Theta$ and the outer loop count reaches the maximum, the algorithm stops by outputting an arbitrary point $\theta \in \Theta$. We clarify the notations as follows.

**Notations.**  Let $\pi$ and $p^*$ be the unconstrained and constrained Gibbs distribution respectively with density $\pi(\theta) \propto e^{-\gamma J(\theta)}$, and $p^*(\theta) \propto e^{-\gamma J(\theta)} \mathbf{1}(\theta \in \Theta)$. Let $\hat{p}^K$ be the distribution of the unconstrained MCMC sample after $K$ inner-loop iterations. Let $p^K(\theta) \propto \hat{p}^K(\theta) \mathbb{1} \{\theta \in \Theta\}$ be the distribution of the accepted MCMC sample. Let $p_{\text{out}}$ be the output distribution of Algorithm 1. Let $p_{\text{Fail}}$ be the distribution within domain $\Theta$, which is the output distribution when none of the $\tau_{\max}$ trials belong to $\Theta$.

**Overview.**  We first bound the error in the inner loop – the TV distance between the unconstrained MCMC distribution and the unconstrained Gibbs distribution: $d_{\text{TV}}(\hat{p}^K, \pi)$. Next, we bound the error of the accepted sample – the TV distance between the constrained MCMC distribution and the target distribution: $d_{\text{TV}}(p^K, p^*)$. Finally, we bound the TV distance between the output distribution and the target distribution: $d_{\text{TV}}(p_{\text{out}}, p^*)$.

For the inner-loop part, we cite the previous results. Consider the MALA algorithm on $\mathbb{R}^d$ with a step size of

$$
h = \boldsymbol{\Theta} \left( \min \left\{ \kappa^{-1/2}(\gamma L)^{-1}(\ln \beta_0/\xi)^{-1/2}, \frac{1}{\gamma L d} \right\} \right),
$$

and an initial distribution $\mu^0$ satisfying $\beta_0$-warmness assumption that $\sup_{A \subset \mathbb{R}^d} \left( \frac{\mu^0(A)}{\pi(A)} \right) \leq \beta_0$. Then by Theorem 1 of Dwivedi et al. (2019), we know that the MALA algorithm with the number of iterations

$$
K = \boldsymbol{\Theta} \left( \ln \left( \frac{\beta_0}{\xi} \right) \max \left\{ \kappa^{3/2} \sqrt{\ln \beta_0/\xi}, d\kappa \right\} \right),
$$

converges to $\xi$ accuracy in terms of the TV distance: $d_{TV}(\hat{p}^K, \pi) \leq \xi$. By Lemma 8 of Ma et al. (2019), we also know that on the unconstrained space, the Gaussian distribution $\mathcal{N}(0, \frac{1}{\gamma L}\mathbb{I})$ is $(2\kappa)^{d/2}$ warm with respect to the posterior distribution $\pi$ with condition number $\kappa$. Plugging in this warmness constant, we obtain that

$$
K = \boldsymbol{\Theta} \left( (d \ln \kappa + \ln 1/\xi) \max \left\{ \kappa^{3/2} \sqrt{d \ln \kappa + \ln 1/\xi}, d\kappa \right\} \right),
$$

Next, for the outer-loop (rejection sampling) part, we note that if the unconstrained $\hat{p}^K$ and $\pi$ satisfies $d_{TV}(\hat{p}^K(\theta), \pi(\theta)) \leq \xi$, then we also obtain a bound for $d_{TV}(p^K(\theta), p^*(\theta))$, the TV distance between the constrained distributions $p^K(\theta) \propto \hat{p}^K(\theta) \mathbb{1} \{\theta \in \Theta\}$ and $p^*(\theta) \propto \pi(\theta) \mathbb{1} \{\theta \in \Theta\}$. Normalizing the density gives that $p^*(\theta) = \frac{\pi(\theta) \mathbb{1} \{\theta \in \Theta\}}{\int \pi(\theta) \mathbb{1} \{\theta \in \Theta\} \mathrm{d}\theta}$. Applying the concentration argument above provides the bound: $\int \pi(\theta) \mathbb{1} \{\theta \in \Theta\} \mathrm{d}\theta \geq 1 - 2 \exp \left( -\frac{\gamma m R_1^2}{32d} \right) \geq \frac{1}{2}$, where the last inequality follows by plugging in our choice

of $R_1$. From the definition of the TV-distance, we know that

$$
\begin{aligned}
&d_{TV}\left(p^K(\theta), p^*(\theta)\right) \\
&= \int_{\theta \in \Theta} |p^K(\theta) - p^*(\theta)| \mathrm{d}\theta \\
&= \int_{\theta \in \Theta} \left| \frac{\hat{p}^K(\theta)}{\int_{\theta \in \Theta} \hat{p}^K(\theta) \mathrm{d}\theta} - \frac{p^*(\theta)}{\int_{\theta \in \Theta} \pi(\theta) \mathrm{d}\theta} \right| \mathrm{d}\theta \\
&\leq \int_{\theta \in \Theta} \left| \frac{\hat{p}^K(\theta)}{\int_{\theta \in \Theta} \hat{p}^K(\theta) \mathrm{d}\theta} - \frac{\hat{p}^K(\theta)}{\int_{\theta \in \Theta} \pi(\theta) \mathrm{d}\theta} \right| \mathrm{d}\theta + \int_{\theta \in \Theta} \left| \frac{\hat{p}^K(\theta)}{\int_{\theta \in \Theta} \pi(\theta) \mathrm{d}\theta} - \frac{p^*(\theta)}{\int_{\theta \in \Theta} \pi(\theta) \mathrm{d}\theta} \right| \mathrm{d}\theta \\
&= \left| 1 - \frac{\int_{\theta \in \Theta} \hat{p}^K(\theta) \mathrm{d}\theta}{\int_{\theta \in \Theta} \pi(\theta) \mathrm{d}\theta} \right| + \frac{1}{\int_{\theta \in \Theta} \pi(\theta) \mathrm{d}\theta} \int_{\theta \in \Theta} |\hat{p}^K(\theta) - \pi(\theta)| \mathrm{d}\theta \\
&\overset{(i)}{\leq} 2 \frac{\xi}{\int_{\theta \in \Theta} \pi(\theta) \mathrm{d}\theta} \\
&\leq 4\xi,
\end{aligned}
$$

where (i) follows from the definition of the TV-distance.

Replacing $\xi$ by $\xi/8$, we have $d_{\mathrm{TV}}(p^K, p^*) \leq \xi/2$.

Then, we bound the "failure" probability – the probability that the outer-loop count reaches $\tau_{\max} = \ln(2/\xi)$, while the MCMC sample still falls outside $\Theta$. For the rejection sampling part, we know that for each individual trial, the rejection probability equals the probability mass outside of $\Theta$. Invoking Lemma 25 below, we obtain that for $\hat{\theta} \sim \pi$,

$$
\mathbb{P}\left( \left\| \hat{\theta} - \mathbb{E}_{\theta \sim p(\theta)}[\theta] \right\| \geq r \right) \leq 2e^{-\frac{r^2 \gamma m}{8d}},
$$

and that for $\theta^*$ being the mode of $\pi$,

$$
\left\| \mathbb{E}_{\theta \sim p(\theta)}[\theta] - \theta^* \right\|^2 \leq \frac{3}{\gamma m}.
$$

On the other hand, $\mathbb{B}(\theta^*, R_1) \subset \Theta$. Since $R_1 \geq 8\sqrt{\frac{d}{\gamma m}} \geq 2\sqrt{\frac{3}{\gamma m}}$, we have $\mathbb{B}(\mathbb{E}_{\theta \sim p(\theta)}[\theta], R_1/2) \subset \mathbb{B}(\theta^*, R_1) \subset \Theta$. Hence, applying the concentration argument above for $\hat{\theta} \sim \pi$,

$$
\mathbb{P}\left( \hat{\theta} \notin \Theta \right) \leq 2e^{-\frac{\gamma m R_1^2}{32d}}.
$$

Combining with the definition of the TV distance, we know that for $\hat{\theta}^K \sim \hat{p}^K$ obtained by running the MALA algorithm (on $\mathbb{R}^d$) for $K$ steps,

$$
\mathbb{P}\left( \hat{\theta}^K \notin \Theta \right) \leq \mathbb{P}\left( \hat{\theta} \notin \Theta \right) + d_{TV}(\hat{p}^K, \pi) \leq 2e^{-\frac{\gamma m R_1^2}{32d}} + \xi.
$$

Hence with $\tau_{\max}$ independent trials, the probability that none of the $\tau_{\max}$ trials belong to $\Theta$ is

$$
\begin{aligned}
\mathbb{P}\left(\text{None of the } \tau_{\max} \text{ trials belong to } \Theta\right) = \left( \mathbb{P}\left( \hat{\theta}^K \notin \Theta \right) \right)^{\tau_{\max}} &\leq \left( 2e^{-\frac{\gamma m R_1^2}{32d}} + \xi \right)^{\tau_{\max}} \\
&\leq \left( 2e^{-2} + \xi \right)^{\tau_{\max}} \\
&\leq e^{-\tau_{\max}},
\end{aligned}
$$

where we plug in $R_1 \geq 8\sqrt{\frac{d}{\gamma m}}$, and without loss of generality assume $\xi < 1/16 < e^{-1} - 2e^{-2}$. Therefore, by setting $\tau_{\max} = \ln(2/\xi)$, we have $\mathbb{P}\left(\text{None of the } \tau_{\max} \text{ trials belong to } \Theta\right) \leq \xi/2$.

Finally, we bound the TV distance between the output distribution and the target distribution: $d_{\mathrm{TV}}(p_{\mathrm{out}}, p^*)$. Denote the event $H_{\mathrm{Fail}} = \{\text{None of the } \tau_{\max} \text{ trials belong to } \Theta\}$, and let $H_{\mathrm{Success}} = H_{\mathrm{Fail}}^c$ be its complement. We know from the above analysis that $\mathbb{P}(H_{\mathrm{Fail}}) \leq \xi/2$.

Let $\theta_{\mathrm{out}} \sim p_{\mathrm{out}}$. Notice that

$$\theta_{\mathrm{out}} | H_{\mathrm{Success}} \sim p^K, \text{ and } \theta_{\mathrm{out}} | H_{\mathrm{Fail}} \sim p_{\mathrm{Fail}}.$$

We thus have $p_{\mathrm{out}} = (1 - \mathbb{P}(H_{\mathrm{Fail}}))p^K + \mathbb{P}(H_{\mathrm{Fail}})p_{\mathrm{Fail}}$. Therefore, for

$$\begin{aligned}
d_{\mathrm{TV}}(p_{\mathrm{out}}, p^*) &= \sup_A |p_{\mathrm{out}}(A) - p^*(A)| \\
&= \sup_A \left|(1 - \mathbb{P}(H_{\mathrm{Fail}}))p^K(A) + \mathbb{P}(H_{\mathrm{Fail}})p_{\mathrm{Fail}}(A) - p^*(A)\right| \\
&\leq \sup_A \left|p^K(A) - p^*(A)\right| + \sup_A \left|-\mathbb{P}(H_{\mathrm{Fail}})p^K(A) + \mathbb{P}(H_{\mathrm{Fail}})p_{\mathrm{Fail}}(A)\right| \\
&= d_{\mathrm{TV}}(p^K, p^*) + \mathbb{P}(H_{\mathrm{Fail}})d_{TV}(p^K, p_{\mathrm{Fail}}) \\
&\leq \xi/2 + \xi/2 = \xi,
\end{aligned}$$

where we apply $d_{\mathrm{TV}}(p^K, p^*) \leq \xi/2$, $\mathbb{P}(H_{\mathrm{Fail}}) \leq \xi/2$, and $d_{TV}(p^K, p_{\mathrm{Fail}}) \leq 1$ in the last inequality.

Therefore, we prove that $d_{\mathrm{TV}}(p_{\mathrm{out}}, p^*) \leq \xi$.

At last, we calculate the expected total number of iterations. Let $\tau_{\mathrm{halt}}$ be the current count of the outer loop when the algorithm stops. Then $\tau_{\mathrm{halt}}$ follows the finite geometric distribution with the probability of success in each trial equals $\mathbb{P}\left(\hat{\theta}^K \in \Theta\right) \geq 1 - 2e^{-2} - \xi$, and the maximum number of trials $\tau_{\max}$. Therefore, $\mathbb{E}(\tau_{\mathrm{halt}}) \leq \frac{1}{1-2e^{-2}-\xi} \leq \frac{1}{1-2e^{-2}-1/8} < 2$, when $\xi < 1/16 < 1/8$.

Therefore, the expected number of total iterations is given by

$$K \cdot \mathbb{E}(\tau_{\mathrm{halt}}) = \mathbf{\Theta}\left((d \ln \kappa + \ln 1/\xi) \max\left\{\kappa^{3/2}\sqrt{d \ln \kappa + \ln 1/\xi}, d\kappa\right\}\right),$$

which completes the proof. ∎

The universal constants regarding the number of iterations can be found in the proof of Lemma 7 in Ma et al. (2019).

**Lemma 25.** *In $\mathbb{R}^d$, if a distribution $p$ is $\gamma m$-strongly log-concave, then*

$$\left\|\mathbb{E}_{\theta \sim p(\theta)}[\theta] - \theta^*\right\| \leq \sqrt{\frac{3}{\gamma m}},$$

*where $\theta^*$ denotes the mode of the distribution $p(\theta)$. We also have that for $\hat{\theta} \sim p$,*

$$\mathbb{P}\left(\left\|\hat{\theta} - \mathbb{E}_{\theta \sim p(\theta)}[\theta]\right\| \geq r\right) \leq 2e^{-\frac{r^2 \gamma m}{8d}}.$$

**Proof** We prove this Lemma following the proof of Lemma 10 in Mazumdar et al. (2020).

For $\left\|\mathbb{E}_{\theta \sim p(\theta)}[\theta] - \theta^*\right\|$, we use the fact that $p(\theta)$ is $\gamma m$-strongly log-concave and consequently unimodal, and that

$$\left\|\mathbb{E}_{\theta \sim p(\theta)}[\theta] - \theta^*\right\|^2 \leq \frac{3}{\gamma m}.$$

For $\left\|\hat{\theta} - \mathbb{E}_{\theta \sim p(\theta)}[\theta]\right\|$, we note that $p(\theta)$ being $\gamma m$-strongly log-concave implies that the random variable $\theta \sim p$ is a sub-Gaussian random vector with parameter $\sigma_v^2 = \frac{1}{2\gamma m}$. Consequently, it is a norm-sub-Gaussian random variable with constant $\sigma_n^2 = \frac{4d}{\gamma m}$. Plugging into the definition of norm-sub-Gaussian random variable, we obtain that for $\hat{\theta} \sim \pi$,

$$\mathbb{P}\left(\left\|\hat{\theta} - \mathbb{E}_{\theta \sim p(\theta)}[\theta]\right\| \geq r\right) \leq 2e^{-\frac{r^2 \gamma m}{8d}}.$$

■

**Proof of Corollary 10** Applying Lemma 8, to obtain $\Delta$-accuracy in $W_\infty$ distance, we want that

$$\xi < p_{\min} \cdot \frac{\pi^{d/2}}{2^{d+1} \cdot \Gamma(\frac{d}{2}+1)d^{d/2}} \Delta^d,$$

where $p_{\min} \leq \min_\theta p^*(\theta)$ and $p^*(\theta) \propto e^{-\gamma J(\theta)} \mathbf{1}(\theta \in \Theta)$ as defined in Section 3.3.

Since $\theta^* \in \Theta$ by Assumption 3, applying Corollary 20, we obtain that

$$\min_\theta p^*(\theta) \geq e^{-2\gamma n \beta B^2} \cdot \frac{\Gamma(\frac{d}{2}+1)}{\pi^{d/2} B^d}.$$

We then set

$$\xi = p_{\min} \cdot \frac{\pi^{d/2}}{2^{d+1} \cdot \Gamma(\frac{d}{2}+1)d^{d/2}} \Delta^d \qquad\qquad (p_{\min} = e^{-2\gamma n \beta B^2} \cdot \frac{\Gamma(\frac{d}{2}+1)}{\pi^{d/2} B^d})$$

$$= e^{-2\gamma n \beta B^2} \cdot \frac{1}{2^{d+1} \cdot d^{d/2} B^d} \Delta^d$$

Therefore, we have

$$\ln 1/\xi \sim \mathcal{O}\left(\gamma n \beta B^2 + d \log d + d \log B + d \log(1/\Delta)\right).$$

Therefore, applying Theorem 1, we obtain that Algorithm 1 converges to $\Delta$-accuracy in $W_\infty(p_{\text{out}}, p^*)$ with the following expected number of gradient queries to $J$:

$$\Theta\left(\left(d \ln(\kappa B d) + d \ln(1/\Delta) + \gamma n \beta B^2\right) \max\left\{\kappa^{3/2}\sqrt{d \ln(\kappa B d) + d \ln(1/\Delta) + \gamma n \beta B^2}, d\kappa\right\}\right).$$

■

## G   PROOF OF THEOREM 2

We first provide the following lemma that converts the distance between probability measures to the distance between random variables.

**Lemma 26.** *Let $P, Q$ be two distributions defined on $\Theta$ such that $W_\infty(P, Q) \leq \Delta$ under a metric* dist : $\Theta \times \Theta \to \mathbb{R}_+$. *Let $X \sim P$, there exists a random variable $Y \sim Q$ such that $\mathbb{P}[\text{dist}(X, Y) \leq \Delta] = 1$.*

The proof of this lemma is deferred to the end of this section.

**Proof of Theorem 2** Let $x \sim \widetilde{\mathcal{M}}(D)$, by Lemma 26 and the condition on $W_\infty$ distance, there is a coupling $\zeta$ such that random variable $y \sim \mathcal{M}(D)$, $(x, y) \sim \zeta$, and $\text{dist}(x, y) \leq \Delta$ with probability 1 (with respect to $\zeta$). The distribution of $x$ is equivalent to the following two-step procedure

(1) $y \sim \mathcal{M}(D)$

(2) $x \sim \zeta(\cdot|y)$

The second step can be further viewed as first sampling $u \sim \text{Uniform}([0,1])$ and then mapping $u$ to $x$ deterministically by a data-dependent function $x = g(u)$ where $g$ is completely determined by $\zeta(\cdot|y)$ via the inverse integral transform. $g$ depends on $D$ and $y$ through $\zeta$.

Define query $f_{u,y}(D) = g_\zeta(u) - y$ and on a neighbor dataset $D'$, $f_{u,y}(D') = g_{\zeta'}(u) - y$ where $\zeta'$ is the resulting coupling under $D'$. With probability 1 under $\zeta$, $\|g_\zeta(u) - y\| \le \Delta$. Similarly with probability 1, under $\zeta'$, $\|g_{\zeta'}(u) - y\| \le \Delta$.

By a union bound, with probability 1 (under $\zeta(\cdot|y) \times \zeta'(\cdot|y)$), the global sensitivity of this query $f_{u,y}$ satisfies
$$\|f_{u,y}(D) - f_{u,y}(D')\| \le \|g_\zeta(u) - y\| + \|g_{\zeta'}(u) - y\| \le 2\Delta.$$
Recall that ASAP returns $\tilde{x} = x + Z = y + (x - y) + Z$. In the above, $x - y = f_{u,y}(D)$. By choosing $Z$ appropriately as the Gaussian mechanism (or Laplace mechanism) and the adaptive composition theorem (of GDP and pure-DP), we establish the two stated claims. ∎

**Proof of Lemma 26** By the attainability of infimum for $W_\infty$ stated in Lemma 22 and that $W_\infty(P, Q) \le \Delta$, we know that there exists a joint distribution $\zeta(x, y)$, such that the marginals in $x$ and $y$ follow $P$ and $Q$ respectively, and that $\text{ess sup}_{(x,y) \in (\Theta \times \Theta, \zeta)} \text{dist}(x, y) = W_\infty(P, Q) \le \Delta$. For given $X \sim P$, we define a conditional distribution $\zeta(\cdot|x)$, then taking $Y \sim \zeta(\cdot|x)$ statisfies $\mathbb{P}[\text{dist}(X, Y) \le \Delta] = 1$.

∎

## H   PROOF OF THEOREM 3

**Proof of Theorem 3** We divide the proof into three parts: privacy, accuracy, and computation. We first present the proof for pure DP case.

**Privacy.** The privacy analysis follows from the adaptive composition (Lemma 14) of output perturbation $\varepsilon$-DP and ASAP ($2\varepsilon$-DP, Theorem 2).

**Accuracy.** Let $E$ denote the event of $\|\theta_0 - \theta_{\text{opt}}\|_1 \le \frac{2d(G + \tau\alpha)\ln(d/\rho)}{n\alpha\varepsilon}$, where $\theta_0, \theta_{\text{opt}}$ is defined in Algorithm 4 for pure DP. Then by Lemma 17 of (Chaudhuri et al., 2011), $\mathbb{P}(E) \ge 1 - \rho$. By the choice of $B$ from (2), we know that under event $E$, we have $\mathbb{B}(\theta^*, R_1) \subset \Theta$, where $R_1 = 8\sqrt{\frac{d}{\gamma\alpha n}}$, and thus Assumption 3 holds.

Denote $\tilde{\theta}$ the output of MALA with constraint (Algorithm 1), and denote $p^*$ the exact posterior.

We then divide the risk into two parts
$$\mathbb{E}\left[\mathcal{L}(\hat{\theta})\right] - \mathcal{L}(\theta^*) = \mathbb{E}\left[\mathcal{L}(\hat{\theta}) - \mathcal{L}(\tilde{\theta})\right] + \mathbb{E}\left[\mathcal{L}(\tilde{\theta})\right] - \mathcal{L}(\theta^*). \tag{5}$$

For the first part, note that $\hat{\theta} = \tilde{\theta} + \mathbf{Z}$, where $Z_i \overset{\text{i.i.d.}}{\sim} \text{Lap}(\Delta/\varepsilon)$, $i = 1, ..., d$, thus we have that

$$\mathbb{E}\left[\|\hat{\theta} - \tilde{\theta}\|_2\right] \overset{\text{Jensen's inequality}}{\le} \sqrt{\mathbb{E}\left[\|\hat{\theta} - \tilde{\theta}\|_2^2\right]} = \sqrt{\mathbb{E}\left[\sum_{i=1}^d Z_i^2\right]} = \sqrt{2d}\frac{\Delta}{\varepsilon}$$

With the $nG$-Lipschitz continuity of $\mathcal{L}$, we have

$$\mathbb{E}\left[\mathcal{L}(\hat{\theta}) - \mathcal{L}(\tilde{\theta})\right] \le nG\mathbb{E}\left[\|\hat{\theta} - \tilde{\theta}\|_2\right] \le \frac{\sqrt{2d}nG\Delta}{\varepsilon} \tag{6}$$

For $\mathbb{E}\left[\mathcal{L}(\tilde{\theta})\right] - \mathcal{L}(\theta^*)$, the second part of the right-hand side of the equation (5), we consider two cases:

1. Under the event $E$. By Corollary 10, we have $W_\infty(\tilde{p}, p^*) \le \Delta$. Since $W_\infty(\tilde{p}, p^*) \le \Delta$, applying Lemma 22, there exist a coupling of $\tilde{p}$ and $p^*$, denoted as $\zeta^*$, such that $\operatorname{ess\,sup}_{(x,y)\in(\Theta\times\Theta,\zeta)}\|x - y\|_1 \le \Delta$, thus $\operatorname{ess\,sup}_{(x,y)\in(\Theta\times\Theta,\zeta)}\|x-y\|_2 \le \Delta$. Take $\theta_{\text{post}}|\tilde{\theta} \sim \zeta(\cdot|\tilde{\theta})$, then we know $\theta_{\text{post}} \sim p^*$.

   We divide

   $$\mathbb{E}\left[\mathcal{L}(\tilde{\theta})|E\right] - \mathcal{L}(\theta^*) = \mathbb{E}\left[\mathcal{L}(\tilde{\theta}) - \mathcal{L}(\theta_{\text{post}})|E\right] + \left(\mathbb{E}\left[\mathcal{L}(\theta_{\text{post}})|E\right] - \mathcal{L}(\theta^*)\right) \tag{7}$$

   With the $nG$-Lipschitz continuity of $\mathcal{L}$, we have

   $$\mathbb{E}_{(\tilde{\theta},\theta_{\text{post}})\sim\zeta}\left[\mathcal{L}(\tilde{\theta}) - \mathcal{L}(\theta_{\text{post}})|E\right] \le nG\mathbb{E}_{(\tilde{\theta},\theta_{\text{post}})\sim\zeta}\left[\|\tilde{\theta} - \theta_{\text{post}}\|_2|E\right] \le nG\Delta, \tag{8}$$

   which is dominated by (6). By Lemma 6, we have

   $$\mathbb{E}[\mathcal{L}(\theta_{\text{post}})|E] - \mathcal{L}(\theta^*) \le \frac{d}{\gamma}. \tag{9}$$

2. Under the complementary event $E^c$. For simplicity, we set $\tau$ in Algorithm 4 to be sufficiently small and exclude this factor. By the $n\beta$-Lipschitz smooth of $\mathcal{L}$ and the first-order condition, we have

   $$\mathbb{E}\left[\mathcal{L}(\tilde{\theta})|E^c\right] - \mathcal{L}(\theta^*) \le \frac{n\beta}{2}\mathbb{E}\left[\|\tilde{\theta} - \theta^*\|_2^2|E^c\right] \le \frac{n\beta}{2}\mathbb{E}\left[\left(\|\tilde{\theta} - \theta_0\|_2 + \|\theta_0 - \theta^*\|_2\right)^2|E^c\right]$$
   $$\le \frac{n\beta}{2}\mathbb{E}\left[\left(B + \|\theta_0 - \theta^*\|_1\right)^2|E^c\right] \tag{10}$$

   By Lemma 27, we have

   $$\mathbb{E}\left[\|\theta_0 - \theta^*\|_1^2|E^c\right] \le \mathcal{O}\left(d(d+1)\left(\frac{\sqrt{d}G}{\alpha n\varepsilon}\right)^2 + B^2 + (d+1)B\frac{\sqrt{d}G}{\alpha n\varepsilon}\right),$$

   and

   $$\mathbb{E}\left[\|\theta_0 - \theta^*\|_1|E^c\right] \le \mathcal{O}\left(d\frac{\sqrt{d}G}{\alpha n\varepsilon} + B\right).$$

   Plugging these into (10), plugging in $\kappa = \beta/\alpha$, we obtain that

   $$\mathbb{E}[\mathcal{L}(\tilde{\theta})|E^c] - \mathcal{L}(\theta^*) \le \mathcal{O}\left(\frac{d^3G^2\kappa}{\alpha n\varepsilon^2}\right). \tag{11}$$

Therefore, adding (6), (8), and (9), as well as adding (6), and (11), respectively, we obtain

$$\mathbb{E}\left[\mathcal{L}(\hat{\theta})|E\right] - \mathcal{L}(\theta^*) \le \frac{\sqrt{2d}nG\Delta}{\varepsilon} + nG\Delta + \frac{d}{\gamma} = \mathcal{O}\left(\frac{d^2G^2\ln(d/\rho)}{\alpha n\varepsilon^2}\right), \text{ and} \tag{12}$$

$$\mathbb{E}\left[\mathcal{L}(\hat{\theta})|E^c\right] - \mathcal{L}(\theta^*) \le \frac{\sqrt{2d}nG\Delta}{\varepsilon} + \mathcal{O}\left(\frac{d^3G^2\kappa}{\alpha n\varepsilon^2}\right) = \mathcal{O}\left(\frac{d^2G^2\left(\ln(d/\rho) + d\kappa\right)}{\alpha n\varepsilon^2}\right), \tag{13}$$

where we instantiated our choice of $\gamma = \frac{\varepsilon}{G\mathrm{Diam}(\Theta_{\text{local}})}$ with $\mathrm{Diam}(\Theta_{\text{local}}) = 2B = \frac{2CdG\log(d/\rho)}{\alpha n\varepsilon}$, and $\Delta = \frac{dG\log(d/\rho)}{2n^2\alpha\varepsilon} \le \frac{d^{\frac{3}{2}}G\log(d/\rho)}{2n^2\alpha\varepsilon}$.

With (12) and (13), we have

$$\mathbb{E}\left[\mathcal{L}(\hat{\theta})\right] - \mathcal{L}(\theta^*) = \mathbb{P}(E) \cdot \mathbb{E}\left[\mathcal{L}(\hat{\theta})|E\right] + (1 - \mathbb{P}(E)) \cdot \mathbb{E}\left[\mathcal{L}(\hat{\theta})|E^c\right] - \mathcal{L}(\theta^*)$$

$$\leq \mathcal{O}\left(\frac{d^2 G^2 \ln(d/\rho)}{\alpha n \varepsilon^2}\right) + \rho \cdot \mathcal{O}\left(\frac{d^2 G^2 \left(\ln(d/\rho) + d\kappa\right)}{\alpha n \varepsilon^2}\right)$$

$$= \mathcal{O}\left(\frac{d^2 G^2 \left(\ln d + \ln(1/\rho) + \rho d\kappa\right)}{\alpha n \varepsilon^2}\right).$$

Taking $\rho = \mathcal{O}\left(1/(d\kappa)\right)$, we obtain

$$\mathbb{E}\left[\mathcal{L}(\hat{\theta})\right] - \mathcal{L}(\theta^*) = \mathcal{O}\left(\frac{d^2 G^2 \left(\ln d + \ln \kappa\right)}{\alpha n \varepsilon^2}\right).$$

**Computation.** The optimization oracle (e.g., Newton's method or Gradient Descent) runs in $\mathcal{O}(n \log n)$ time for getting a solution satisfying $\|\theta_0 - \theta^*\| \leq 1/n^2$. For the sampling step, following the proof of Corollary 10, the expected computation complexity is bounded by

$$K \cdot \left((1 - \rho)\mathbb{E}\left[\tau_{\text{halt}}|E\right] + \rho \tau_{\max}\right).$$

Applying Corollary 10 with taking proper $B = \frac{CdG\log(d/\rho)}{\alpha n \varepsilon}$, we have $\mathbb{E}\left[\tau_{\text{halt}}|E\right] \sim \mathcal{O}(1)$.

Plugging $B, \gamma$ as given in Table 2 for pure DP, we obtain

$$\xi = e^{-2\gamma n \beta B^2} \cdot \frac{1}{2^{d+1} \cdot d^{d/2} B^d} \Delta^d$$

$$= e^{-\frac{\varepsilon n \beta B}{G}} \cdot \frac{1}{2^{d+1} \cdot d^{d/2} B^d} \Delta^d \qquad\qquad (\gamma = \frac{\varepsilon}{2GB})$$

$$= e^{-C_1 \kappa d \ln(d/\rho)} \cdot \frac{1}{2^{d+1} \cdot d^{d/2}} \left(\frac{\alpha n \varepsilon}{C_1 G d \ln(d/\rho)}\right)^d \Delta^d \qquad (B = \frac{C_1 G d \ln(d/\rho)}{\alpha n \varepsilon}, \ \kappa = \frac{\beta}{\alpha})$$

Therefore, we have

$$\ln 1/\xi \sim \mathcal{O}\left(\kappa d \ln(d/\rho) + d \ln d + d \ln\left(\frac{Gd\ln(d/\rho)}{\alpha n \varepsilon}\right) + d \ln \frac{1}{\Delta}\right)$$

$$\sim \mathcal{O}\left(\kappa d \ln(d/\rho) + d \ln\left(\frac{Gd\ln(d/\rho)}{\alpha n \varepsilon \Delta}\right)\right) \qquad (\kappa = \frac{\beta}{\alpha} > 1, \ \rho < 1)$$

Therefore, applying Theorem 1, we obtain that Algorithm 1 converges to $\Delta$-accuracy in $W_\infty(p_{\text{out}}, p^*)$ with the following expected number of gradient queries to $J$:

$$\Theta\left(d\left(\ln \kappa + \kappa \ln(d/\rho) + \ln\left(\frac{Gd\ln(d/\rho)}{\alpha n \varepsilon \Delta}\right)\right) \max\left\{\kappa^{3/2}\sqrt{d\left(\kappa \ln(d/\rho) + \ln\left(\frac{Gd\ln(d/\rho)}{\alpha n \varepsilon \Delta}\right)\right)}, d\kappa\right\}\right)$$

$$\sim \Theta\left(d\left(\kappa \ln(d/\rho) + \ln\left(\frac{G\ln(d/\rho)}{\alpha n \varepsilon \Delta}\right)\right) \max\left\{\kappa^{3/2}\sqrt{d\left(\kappa \ln(d/\rho) + \ln\left(\frac{G\ln(d/\rho)}{\alpha n \varepsilon \Delta}\right)\right)}, d\kappa\right\}\right)$$

Plugging in $\Delta = \frac{Gd \log(d/\rho)}{2n^2 \alpha \varepsilon}$ as given in Table 2, we have

$$K \sim \mathbf{\Theta}\left(d\left(\kappa \ln(d/\rho) + \ln n\right) \max\left\{\kappa^{3/2}\sqrt{d\left(\kappa \ln(d/\rho) + \ln n\right)}, d\kappa\right\}\right).$$

Therefore, the expected total computation complexity is given by

$$\mathbf{\Theta}\left(nd\left(\kappa \ln(d/\rho) + \ln n\right) \max\left\{\kappa^{3/2}\sqrt{d\left(\kappa \ln(d/\rho) + \ln n\right)}, d\kappa\right\}\left(1 + \rho d\left(\kappa \ln(d/\rho) + \ln n\right)\right)\right).$$

Taking $\rho = \mathcal{O}\left(1/(d\kappa)\right)$, assuming $(\ln n)/\kappa \leq \mathcal{O}(1)$, it translates into

$$\mathbf{\Theta}\left(nd\left(\kappa \ln(d\kappa) + \ln n\right) \max\left\{\kappa^{3/2}\sqrt{d\left(\kappa \ln(d\kappa) + \ln n\right)}, d\kappa\right\} \ln(d\kappa)\right).$$

**Gaussain DP case.** The analysis of Gaussian DP closely parallels the proof of pure DP case except for the slight difference in the accuracy and computation analysis. Here, we present a simplified proof.

Let $E$ denote the event of $\|\theta_0 - \theta_{\mathrm{opt}}\|_2 \leq \frac{2\sqrt{2}(\tau\alpha + G)(\sqrt{d} + \sqrt{\ln(1/\rho)})}{\alpha n \mu}$, where $\theta_0, \theta_{\mathrm{opt}}$ is defined in Algorithm 4 for Gaussian DP. Since $\theta_0 = \theta_{\mathrm{opt}} + \mathbf{Z}$, where $Z_i \sim \mathcal{N}(0, 4(\tau + \frac{G}{\alpha})^2/n^2\mu^2)$, applying the tail bound for chi-square distributions by Lemma 1 of (Laurent & Massart, 2000), we have $\mathbb{P}(E) \geq 1 - \rho$. By the choice of $B$ from (3), we know that under event $E$, Assumption 3 holds. For simplicity, we set $\tau$ in Algorithm 4 to be sufficiently small and exclude this factor.

Under this event $E$, by Lemma 6, similar to the pure DP case, we have

$$\mathbb{E}[\mathcal{L}(\hat{\theta})|E] - \mathcal{L}(\theta^*) \leq \frac{d}{\gamma} + \mathcal{O}\left(\frac{nG\sqrt{d}\Delta}{\mu}\right) = \mathcal{O}\left(\frac{dG^2}{\alpha n \mu^2}\right),$$

where we instantiated our choice of $\gamma = \frac{\mu^2 \alpha n}{G^2}$, and $\Delta = \frac{\sqrt{d}G}{\sqrt{2}n^2 \alpha \mu}$.

On the other hand, under $E^c$, applying Lemma 27 by taking $k = d/2$, and $\sigma = 8(\frac{\Delta}{\mu})^2$, we have

$$\mathbb{E}\left[\|\theta_0 - \theta_{\mathrm{opt}}\|_2^2|E^c\right] \leq \mathcal{O}\left(\frac{G^2(d + \ln(1/\rho))}{(\alpha n \mu)^2}\right).$$

Thus by the $n\beta$-Lipschitz smooth of $\mathcal{L}$ and the first-order condition, we have

$$\mathbb{E}\left[\mathcal{L}(\tilde{\theta})|E^c\right] - \mathcal{L}(\theta^*) \leq \frac{n\beta}{2}\mathbb{E}\left[\|\tilde{\theta} - \theta^*\|_2^2|E^c\right] \leq \frac{n\beta}{2}\mathbb{E}\left[\left(\|\tilde{\theta} - \theta_0\|_2 + \|\theta_0 - \theta^*\|_2\right)^2|E^c\right]$$
$$\leq \mathcal{O}\left(\frac{\kappa G^2(d + \ln(1/\rho))}{\alpha n \mu^2}\right). \tag{14}$$

Therefore, by combining the two cases, we have

$$\mathbb{E}[\mathcal{L}(\hat{\theta})] - \mathcal{L}(\theta^*) \leq \mathcal{O}\left(\frac{G^2\left(d + \rho\kappa d + \rho\kappa \ln(1/\rho)\right)}{\alpha n \mu^2}\right).$$

Taking $\rho = \mathcal{O}(1/d\kappa)$, we obtain that

$$\mathbb{E}[\mathcal{L}(\hat{\theta})] - \mathcal{L}(\theta^*) \leq \mathcal{O}\left(\frac{G^2(d + \ln\kappa)}{\alpha n \mu^2}\right).$$

Applying Lemma 8, taking the $B, \gamma, \Delta, \rho$ as Table 2 for Gaussian DP, assuming $(\ln n)/\kappa \leq \mathcal{O}(1)$, the expected total computation complexity of the Gaussian DP case is given by

$$\mathbf{\Theta}\left(n\left(d\kappa + d\ln n + \kappa \ln\kappa\right) \max\left\{\kappa^{3/2}\sqrt{d\kappa + d\ln n + \kappa \ln\kappa}, d\kappa\right\} \ln(\kappa)\right).$$

$\blacksquare$

**Lemma 27.** *Let $X$ be a random variable drawn from the Gamma distribution $\Gamma(k, \sigma)$, with the density*

$$f(x; k, \sigma) = \frac{x^{k-1}e^{-\frac{x}{\sigma}}}{\sigma^k \Gamma(k)}, \quad \text{for } x > 0, k > 0, \text{ and } \sigma > 0.$$

*Suppose $k \geq 1$, then for any $a > 0$, we have*

$$\mathbb{E}(X|X > a) \leq k\sigma + a, \quad \text{and} \quad \mathbb{E}(X^2|X > a) \leq k(k+1)\sigma^2 + a^2 + (k+1)a\sigma.$$

**Proof** We have

$$\mathbb{E}(X|X > a) = \frac{\int_a^\infty x f(x; k, \sigma) dx}{\int_a^\infty f(x; k, \sigma) dx} = \frac{\int_a^\infty x^k e^{-\frac{x}{\sigma}} dx}{\int_a^\infty x^{k-1} e^{-\frac{x}{\sigma}} dx}$$

$$= \sigma \frac{\int_{a/\sigma}^\infty x^k e^{-x} dx}{\int_{a/\sigma}^\infty x^{k-1} e^{-x} dx}$$

$$= \sigma \frac{\Gamma(k+1, a/\sigma)}{\Gamma(k, a/\sigma)},$$

where $\Gamma(k, t)$ is the incomplete gamma function defined as $\Gamma(k, t) = \int_t^\infty x^{k-1}e^{-x}\mathrm{d}x$.

Applying integral by parts, we have

$$\Gamma(k+1, t) = k\Gamma(k, t) + t^k e^{-t}.$$

We also have, for $k \geq 1$,

$$\Gamma(k, t) = (k-1)\Gamma(k-1, t) + t^{k-1}e^{-t} \geq t^{k-1}e^{-t}.$$

Therefore, for $k \geq 1$,

$$\frac{\Gamma(k+1, t)}{\Gamma(k, t)} = k + \frac{t^k e^{-t}}{\Gamma(k, t)} \leq k + \frac{t^k e^{-t}}{t^{k-1}e^{-t}} = k + t.$$

Taking $t = a/\sigma$, we have

$$\mathbb{E}(X|X > a) \leq \sigma \frac{\Gamma(k+1, a/\sigma)}{\Gamma(k, a/\sigma)} \leq k\sigma + a.$$

For $\mathbb{E}(X^2|X > a)$, we have

$$\mathbb{E}(X^2|X > a) = \frac{\int_a^\infty x^2 f(x; k, \sigma) dx}{\int_a^\infty f(x; k, \sigma) dx} = \frac{\int_a^\infty x^{k+1} e^{-\frac{x}{\sigma}} dx}{\int_a^\infty x^{k-1} e^{-\frac{x}{\sigma}} dx}$$

$$= \sigma^2 \frac{\int_{a/\sigma}^\infty x^{k+1} e^{-x} dx}{\int_{a/\sigma}^\infty x^{k-1} e^{-x} dx}$$

$$= \sigma^2 \frac{\Gamma(k+2, a/\sigma)}{\Gamma(k, a/\sigma)}$$

$$= \sigma^2 \frac{(k+1)\Gamma(k+1, a/\sigma) + (a/\sigma)^{k+1}e^{-a/\sigma}}{\Gamma(k, a/\sigma)}$$

$$\leq \sigma^2 \left( (k+1)(k + a/\sigma) + \frac{(a/\sigma)^{k+1}e^{-a/\sigma}}{(a/\sigma)^{k-1}e^{-a/\sigma}} \right)$$

$$= \sigma^2 k(k+1) + a^2 + a\sigma(k+1).$$

∎

# I  DEFERRED PROOFS OF THE SUPPORTING LEMMAS AND COROLLARIES

## I.1  PROOF OF LEMMA 9 AND COROLLARY 20

**Proof of Lemma 9**  Denote $\theta^*$ as the minimizer of $J$ on $\Theta$. Then

$$Z = \int_\Theta e^{-\gamma J(\theta)} \mathrm{d}\theta \le e^{-\gamma J(\theta^*)} \int_\Theta \mathrm{d}\theta = e^{-\gamma J(\theta^*)} \cdot \frac{\pi^{d/2}(R/2)^d}{\Gamma(d/2+1)}.$$

On the other hand, by the Lipschitz continuity assumption, $J(\theta) - J(\theta^*) \le G \cdot \|\theta - \theta^*\| \le GR$. Hence

$$\frac{1}{Z} e^{-\gamma J(\theta)} \ge e^{-\gamma GR} \cdot \frac{\Gamma(d/2+1)}{\pi^{d/2}(R/2)^d}.$$

If $J$ instead satisfies the $\beta$-Lipschitz smoothness and convextiy assumptions, then for all $\tilde{\theta} \in \Theta$

$$\begin{aligned}
J(\theta) - J(\theta^*) &= J(\theta) - J(\tilde{\theta}) + J(\tilde{\theta}) - J(\theta^*) \\
&\le \langle J(\tilde{\theta}), \theta - \tilde{\theta} \rangle + \frac{\beta}{2}\|\theta - \tilde{\theta}\|^2 - \langle \nabla J(\tilde{\theta}), \theta^* - \tilde{\theta} \rangle \\
&= \langle J(\tilde{\theta}), \theta - \theta^* \rangle + \frac{\beta}{2}\|\theta - \tilde{\theta}\|^2 \\
&\le \|J(\tilde{\theta})\| R + \frac{1}{2}\beta R^2
\end{aligned}$$

Hence

$$\inf_{\theta \in \Theta} p(\theta) \ge e^{-\gamma\left(R\|\nabla J(\tilde{\theta})\| + \beta R^2/2\right)} \cdot \frac{\Gamma(\frac{d}{2}+1)}{\pi^{d/2}(R/2)^d}, \ \forall \tilde{\beta} \in \Theta.$$

Furthermore, if $\theta^*$ is the global minimizer, i.e., $\nabla J(\theta^*) = 0$, then

$$\inf_{\theta \in \Theta} p(\theta) \ge e^{-\gamma \beta R^2/2} \cdot \frac{\Gamma(\frac{d}{2}+1)}{\pi^{d/2}(R/2)^d}, \ \forall \tilde{\theta} \in \Theta.$$

∎

**Proof of Corollary 20**  It suffices to invoke Lemma 9 by setting $J(\theta) := \sum_{i=1}^n \ell_i(\theta) + \frac{\lambda}{2}\|\theta - \theta_0\|^2$ and $\Theta := \{\theta | \|\theta - \theta_0\| \le B\}$, notice that the diameter of $\Theta$ is $2B$. ∎

## I.2  PROOF OF LEMMA 18 AND LEMMA 19

**Proof of Lemma 18**  By the $\alpha$-strong convexity of $\mathcal{L}_D$ and first-order optimality conditions for $\theta^*(D)$

$$\begin{aligned}
\mathcal{L}_D(\theta^*(D')) &\ge \mathcal{L}_D(\theta^*(D)) + \langle \theta^*(D') - \theta^*(D), \nabla\mathcal{L}_D(\theta^*(D)) \rangle + \frac{\alpha n}{2}\|\theta^*(D) - \theta^*(D')\|^2 \\
&\ge \mathcal{L}_D(\theta^*(D)) + \frac{\alpha n}{2}\|\theta^*(D) - \theta^*(D')\|^2.
\end{aligned}$$

On the other side, by the $\alpha$-strong convexity of $\mathcal{L}_{D'}$ and first-order optimality conditions for $\theta^*(D')$

$$\begin{aligned}
\mathcal{L}_{D'}(\theta^*(D)) &\ge \mathcal{L}_{D'}(\theta^*(D')) + \langle \theta^*(D) - \theta^*(D'), \nabla\mathcal{L}_{D'}(\theta^*(D')) \rangle + \frac{\alpha n}{2}\|\theta^*(D') - \theta^*(D)\|^2 \\
&\ge \mathcal{L}_{D'}(\theta^*(D')) + \frac{\alpha n}{2}\|\theta^*(D) - \theta^*(D')\|^2.
\end{aligned}$$

Add the two inequalities we get

$$\begin{aligned}
\alpha n\|\theta^*(D) - \theta^*(D')\|^2 &\le \mathcal{L}_D(\theta^*(D')) - \mathcal{L}_{D'}(\theta^*(D')) + \mathcal{L}_{D'}(\theta^*(D)) - \mathcal{L}_D(\theta^*(D)) \\
&\le |\ell_x(\theta^*(D')) - \ell_x(\theta^*(D))| \le G\|\theta^*(D) - \theta^*(D')\|.
\end{aligned}$$

The proof is complete by dividing both sides by $\|\theta^*(D) - \theta^*(D')\|$. ∎

**Proof of Lemma 19** It suffices to show the sensitivity of $\tilde{\theta}(D)$ is bounded by $\tilde{\Delta}$. Applying Lemma 18, the sensitivity of $\tilde{\theta}(D)$ is bounded by

$$
\begin{aligned}
\max_{D \simeq D'} ||\tilde{\theta}(D) - \tilde{\theta}(D')||_2 &= \max_{D \simeq D'} ||\tilde{\theta}(D) - \theta^*(D) + \theta^*(D) - \theta^*(D') + \theta^*(D') - \tilde{\theta}(D')||_2 \\
&\leq ||\tilde{\theta}(D) - \theta^*(D)||_2 + \max_{D \simeq D'} ||\theta^*(D) - \theta^*(D')||_2 + ||\theta^*(D') - \tilde{\theta}(D')||_2 \\
&\leq \frac{2\tau}{n} + \frac{2G}{\alpha n} = \tilde{\Delta}.
\end{aligned}
$$

∎

### I.3 PROOF OF LEMMA 14

**Proof of Lemma 14** First, observe that pure-DP (with any $\varepsilon < \infty$) by definition implies absolute continuity. Let $\mu, \nu$ be the two measures induced by a pure-DP mechanism on dataset $D, D'$ respectively. DP implies that for any measurable set $S$, $\mu(S) \leq e^\varepsilon \nu(S)$. This inequality implies that if $\nu(S) = 0$ then $\mu(S) = 0$, which verifies the definition of absolute continuity, i.e., $\mu \ll \nu$.

By our assumption, $\mathcal{M}_1$ satisfies DP, thus $P_{\mathcal{M}_1(D)}$ absolutely continuous w.r.t. $P_{\mathcal{M}_1(D')}$. Similarly, $\mathcal{M}_2(o_1, D)$ satisfies DP for all $o_1$ except when $o_1$ belongs to a measure 0 set, thus with probability 1, $P_{\mathcal{M}_2(O_1, D)}$ is absolutely continuous w.r.t. $P_{\mathcal{M}_2(O_1, D')}$. It follows that the "density" function (technically, Radon-Nikodym derivative) $\frac{dP_{\mathcal{M}_1(D)}}{dP_{\mathcal{M}_1(D')}}$ exists and $\frac{dP_{\mathcal{M}_2(O_1, D)}}{dP_{\mathcal{M}_2(O_1, D')}}$ exists almost surely. In addition, by taking $S = \{\frac{dP_{\mathcal{M}_1(D)}}{dP_{\mathcal{M}_1(D')}} > e^\varepsilon\}$ the DP definition, by a proof by contradiction[*], we have

$$
\mathbb{P}_{O_1 \sim \mathcal{M}_1(D')}\left[\frac{dP_{\mathcal{M}_1(D)}}{dP_{\mathcal{M}_1(D')}}(O_1) \leq e^{\varepsilon_1}\right] = 1 \tag{15}
$$

and

$$
\mathbb{P}_{O_2 \sim \mathcal{M}_2(o_1, D')}\left[\frac{dP_{\mathcal{M}_2(o_1, D)}}{dP_{\mathcal{M}_2(o_1, D')}}(O_2) \leq e^{\varepsilon_2}\right] = 1 \tag{16}
$$

almost surely under $o_1 \sim \mathcal{M}_1(D')$.

Let $O_1 \sim \mathcal{M}_1(D)$ and $O_2 \sim \mathcal{M}_2(O_1, D)$. Similarly, let $\tilde{O}_1 \sim \mathcal{M}_1(D')$ and $\tilde{O}_2 \sim \mathcal{M}_2(\tilde{O}_2, D')$. Consider any measurable set $S \subset \Theta_1 \times \Theta_2$, by the Lebesgue integral

$$
\begin{aligned}
&\mathbb{P}[(O_1, O_2) \in S] \\
&= \int \int \mathbf{1}((u_1, u_2) \in S) dP_{\mathcal{M}_2(u_1, D)}(u_2) dP_{\mathcal{M}_1(D)}(u_1) \\
&= \int \int \mathbf{1}((u_1, u_2) \in S) \frac{dP_{\mathcal{M}_2(u_1, D)}}{dP_{\mathcal{M}_2(u_1, D')}}(u_2) \cdot dP_{\mathcal{M}_2(u_1, D')}(u_2) \cdot \frac{dP_{\mathcal{M}_1(D)}}{dP_{\mathcal{M}_1(D')}}(u_1) \cdot dP_{\mathcal{M}_1(D')}(u_1) \\
&\leq \int \int \mathbf{1}((u_1, u_2) \in S) e^{\varepsilon_2} \cdot dP_{\mathcal{M}_2(u_1, D')}(u_2) \cdot e^{\varepsilon_1} \cdot dP_{\mathcal{M}_1(D')}(u_1) \\
&= e^{\varepsilon_1 + \varepsilon_2} \mathbb{P}[(\tilde{O}_1, \tilde{O}_2) \in S].
\end{aligned}
$$

The inequality above follows from equation 15 and equation 16.

For Gaussian DP, let $P_1$ and $Q_1$ be the probability measures of $M_1(D)$ and $M_1(D')$ respectively, and $P_2(\cdot | x = o)$ and $Q_2(\cdot | x = o)$ be the probability measures of $M_2(o, D)$ and $M_2(o, D')$ respectively. Let $P$ and $Q$ be the probability measures of $M(D)$ and $M(D')$.

---

[*] Assume $S$ is not measure 0. By definition of DP $\mathbb{P}[\mathcal{M}_1(D') \in S] \leq e^\varepsilon \mathbb{P}[\mathcal{M}_1(D') \in S]$ which contradicts with the definition of $S$ unless $S$ has measure 0.

We first show that $P$ is absolutely continuous w.r.t $Q$. Noting that by the definition of Gaussian DP and Hockey-stick divergence, $P_1$ is absolutely continuous w.r.t. $Q_1$ and $P_2(\cdot|x = o)$ is continuous w.r.t. $Q_2(\cdot|x = o)$ with probability 1. Then, for arbitrary $Q$-measureable set $A$, we have that

$$
\begin{aligned}
P(A) &= \int \int \mathbf{1}\{(x, y) \in A\} dP(x, y) \\
&= \int \int \mathbf{1}\{(x, y) \in A\} dP_2(y|x) dP_1(x) \\
&= \int \int \mathbf{1}\{(x, y) \in A\} dP_2(y|x) dP_1(x) \\
&= \int \int \mathbf{1}\{(x, y) \in A\} \frac{dP_2}{dQ_2}(y|x) dQ_2(y|x) \frac{dP_1}{dQ_1}(x) dQ_1(x) \\
&= \int \int \mathbf{1}\{(x, y) \in A\} \frac{dP_2}{dQ_2}(y|x) \frac{dP_1}{dQ_1}(x) dQ(x, y)
\end{aligned}
$$

So $Q(A) = 0$ implies $P(A) = 0$, thus $P$ is absolutely continuous w.r.t $Q$. Moreover, from the above equity, we know that the Radon-Nikodym derivative $\frac{dP}{dQ}(x, y) = \frac{dP_2}{dQ_2}(y|x) \cdot \frac{dP_1}{dQ_1}(x)$.

Since P is absolutely continuous w.r.t Q, their Hockey-stick distance can be defined. We then have for $\alpha > 0$,

$$
\begin{aligned}
&H_\alpha(M(D) \| M(D')) \\
&= H_\alpha(P \| Q) \\
&= \int \int \left[ \frac{dP}{dQ}(x, y) - \alpha \right]_+ dQ(x, y) \\
&= \int \int \mathbf{1}\left\{ \frac{dP}{dQ}(x, y) \geq \alpha \right\} \left( \frac{dP}{dQ}(x, y) - \alpha \right) dQ(x, y) \\
&= \int \int \mathbf{1}\left\{ \frac{dP_2}{dQ_2}(y|x) \cdot \frac{dP_1}{dQ_1}(x) \geq \alpha \right\} \left( \frac{dP_2}{dQ_2}(y|x) \cdot \frac{dP_1}{dQ_1}(x) - \alpha \right) dQ_2(y|x) dQ_1(x) \\
&= \int \int \mathbf{1}\left\{ \frac{dP_2}{dQ_2}(y|x) \cdot \frac{dP_1}{dQ_1}(x) \geq \alpha \right\} \mathbf{1}\left\{ \frac{dP_1}{dQ_1}(x) > 0 \right\} \left( \frac{dP_2}{dQ_2}(y|x) \cdot \frac{dP_1}{dQ_1}(x) - \alpha \right) dQ_2(y|x) dQ_1(x) \\
&= \int \int \mathbf{1}\left\{ \frac{dP_2}{dQ_2}(y|x) \cdot \frac{dP_1}{dQ_1}(x) \geq \alpha \left( \frac{dP_1}{dQ_1}(x) \right)^{-1} \right\} \left( \frac{dP_2}{dQ_2}(y|x) - \alpha \left( \frac{dP_1}{dQ_1}(x) \right)^{-1} \right) dQ_2(y|x) \\
&\quad \cdot \mathbf{1}\left\{ \frac{dP_1}{dQ_1}(x) > 0 \right\} \frac{dP_1}{dQ_1}(x) dQ_1(x) \\
&= \int H_{\alpha\left( \frac{dP_1}{dQ_1}(x) \right)^{-1}} (dP_2(\cdot|x) \| dQ_2(\cdot|x)) \mathbf{1}\left\{ \frac{dP_1}{dQ_1}(x) > 0 \right\} \frac{dP_1}{dQ_1}(x) dQ_1(x) \\
&\leq \int H_{\alpha\left( \frac{dP_1}{dQ_1}(x) \right)^{-1}} (\mathcal{N}(0, 1) \| \mathcal{N}(\mu_2, 1)) \mathbf{1}\left\{ \frac{dP_1}{dQ_1}(x) > 0 \right\} \frac{dP_1}{dQ_1}(x) dQ_1(x) \\
&= \int \int \left[ \frac{dP_{\mathcal{N}(0,1)}}{dP_{\mathcal{N}(\mu_2,1)}}(z) - \alpha \left( \frac{dP_1}{dQ_1}(x) \right)^{-1} \right]_+ dP_{\mathcal{N}(\mu_2,1)}(z) \mathbf{1}\left\{ \frac{dP_1}{dQ_1}(x) > 0 \right\} \frac{dP_1}{dQ_1}(x) dQ_1(x) \\
&= \int \int \left[ \frac{dP_{\mathcal{N}(0,1)}}{dP_{\mathcal{N}(\mu_2,1)}}(z) \frac{dP_1}{dQ_1}(x) - \alpha \right]_+ dP_{\mathcal{N}(\mu_2,1)}(z) dQ_1(x) \\
&= H_\alpha \left( P_1 \times N(0, 1) \| Q_1 \times \mathcal{N}(\mu_2, 1) \right)
\end{aligned}
$$

Continuing this argument, we have

$$H_\alpha\left(P_1 \times N(0,1) \parallel Q_1 \times \mathcal{N}(\mu_2, 1)\right)$$

$$= \int \int \left[\frac{dP_1}{dQ_1}(x) - \alpha \cdot \frac{dP_{\mathcal{N}(\mu_2,1)}}{dP_{\mathcal{N}(0,1)}}(z)\right]_+ \frac{dP_{\mathcal{N}(0,1)}}{dP_{\mathcal{N}(\mu_2,1)}}(z) dQ_1(x) dP_{\mathcal{N}(\mu_2,1)}(z)$$

$$= \int H_{\alpha \cdot \frac{dP_{\mathcal{N}(\mu_2,1)}}{dP_{\mathcal{N}(0,1)}}(z)}(P_1 \| Q_1) \frac{dP_{\mathcal{N}(0,1)}}{dP_{\mathcal{N}(\mu_2,1)}}(z) dP_{\mathcal{N}(\mu_2,1)}(z)$$

$$\leq \int H_{\alpha \cdot \frac{dP_{\mathcal{N}(\mu_2,1)}}{dP_{\mathcal{N}(0,1)}}(z)}(\mathcal{N}(0,1) \| \mathcal{N}(\mu_1, 1)) \frac{dP_{\mathcal{N}(0,1)}}{dP_{\mathcal{N}(\mu_2,1)}}(z) dP_{\mathcal{N}(\mu_2,1)}(z)$$

$$= \int \int \left[\frac{dP_{\mathcal{N}(0,1)}}{dP_{\mathcal{N}(\mu_1,1)}}(w) - \alpha \cdot \frac{dP_{\mathcal{N}(\mu_2,1)}}{dP_{\mathcal{N}(0,1)}}(z)\right]_+ \frac{dP_{\mathcal{N}(0,1)}}{dP_{\mathcal{N}(\mu_2,1)}}(z) dP_{\mathcal{N}(\mu_1,1)}(w) dP_{\mathcal{N}(\mu_2,1)}(z)$$

$$= \int \int \left[\frac{dP_{\mathcal{N}(0,1)}}{dP_{\mathcal{N}(\mu_1,1)}}(w) \frac{dP_{\mathcal{N}(0,1)}}{dP_{\mathcal{N}(\mu_2,1)}}(z) - \alpha\right]_+ dP_{\mathcal{N}(\mu_1,1)}(w) dP_{\mathcal{N}(\mu_2,1)}(z)$$

$$= H_\alpha\left(\mathcal{N}(0,1) \times N(0,1) \| \mathcal{N}(\mu_1, 1) \times \mathcal{N}(\mu_2, 1)\right)$$

By taking $s = (\mu_1 w + \mu_2 z)/\sqrt{\mu_1^2 + \mu_2^2}$, $t = (\mu_1 w - \mu_2 z)/\sqrt{\mu_1^2 + \mu_2^2}$, we have,

$$H_\alpha\left(\mathcal{N}(0,1) \times N(0,1) \| \mathcal{N}(\mu_1, 1) \times \mathcal{N}(\mu_2, 1)\right)$$

$$= \int \int \left[\frac{\exp(-(w^2 + z^2))}{\exp(-((w-\mu_1)^2 + (z-\mu_2)^2))} - \alpha\right]_+ \frac{1}{2\pi} \exp(-((w-\mu_1)^2 + (z-\mu_2)^2)) dw dz$$

$$= \int \int \left[\exp(-2(\mu_1 w + \mu_2 z) + \mu_1^2 + \mu_2^2) - \alpha\right]_+ \frac{1}{2\pi} \exp(-(w^2 + z^2 - 2(\mu_1 w + \mu_2 z) + \mu_1^2 + \mu_2^2)) dw dz$$

$$= \int \int \left[\exp(-2s\sqrt{\mu_1^2 + \mu_2^2} + \mu_1^2 + \mu_2^2) - \alpha\right]_+ \frac{1}{2\pi} \exp(-(s^2 + t^2 - 2s\sqrt{\mu_1^2 + \mu_2^2} + \mu_1^2 + \mu_2^2)) dt ds$$

$$= \int \int \left[\exp(-2s\sqrt{\mu_1^2 + \mu_2^2} + \mu_1^2 + \mu_2^2) - \alpha\right]_+ \frac{1}{\sqrt{2\pi}} \exp(-(s^2 - 2s\sqrt{\mu_1^2 + \mu_2^2} + \mu_1^2 + \mu_2^2)) ds$$

$$= H_\alpha\left(\mathcal{N}(0,1) \| \mathcal{N}(\sqrt{\mu_1^2 + \mu_2^2}, 1)\right)$$

The proof is completed. ∎

### I.4   PROOF OF LEMMA 5

**Proof of Lemma 5** (Dong et al., 2020) showed that exponential sampling with utility function $q(D, \theta)$ that satisfies a property called *bounded range* is differentially private even if the sensitivity is not bounded (unlike the original exponential mechanism).

$$\text{range}(q) := \sup_{D, D' \text{ are neighbors}} (\max_{\theta \in \Theta} - \min_{\theta \in \Theta})[q(D, \theta) - q(D', \theta)].$$

In our problem, $q$ is the (regularized) sum of loss functions, and the difference in $q$ between two neighbor datasets created by adding or removing a datapoint is simply the loss of one data point. $q(D, \theta) - q(D', \theta) = \pm \ell(\theta)$. It is easy to see that the $\text{range}(q) \leq G\text{Diam}(\Theta)$. By (Dong et al., 2020), we get that choosing $\gamma \leq \frac{\varepsilon}{G\text{Diam}(\Theta)}$ gives $\varepsilon$-DP. ∎

### I.5   PROOF OF LEMMA 15

**Proof** Denote

$$f_{\varepsilon,0}(x) = \max\{0, 1 - e^\varepsilon x, e^{-\varepsilon}(1-x)\}, \text{ for } 0 \leq x \leq 1,$$

and
$$G_\mu(x) = \Phi\left(\Phi^{-1}(1-x) - \mu\right), \text{ for } 0 \leq x \leq 1.$$
By Definition 3, Proposition 3 and Definition 4 of Dong et al. (2022), it suffices to show that
$$f_{\varepsilon,0}(x) \geq G_\mu(x), \text{ for all } 0 \leq x \leq 1.$$
By the concavity of $G_\mu$ and the piece-wise linearity of $f_{\varepsilon,0}$, it suffices to show that $G_\mu(x_0) \leq f_{\varepsilon,0}(x_0)$, with $x_0 = \frac{1}{1+e^\varepsilon}$ satisfying $1 - e^\varepsilon x_0 = e^{-\varepsilon}(1 - x_0) = x_0$. Taking $\mu = 2\Phi^{-1}\left(\frac{e^\varepsilon}{1+e^\varepsilon}\right)$, we have $G_\mu(x_0) = f_{\varepsilon,0}(x_0)$, which finishes the proof. ∎

### I.6 Proof of Lemma 22

**Proof of Lemma 22** Denote $\|\text{dist}(\cdot,\cdot)\|_{L_\infty(\Theta^2, \zeta)} = \text{ess sup}_{(x,y) \in (\Theta \times \Theta, \zeta)} \text{dist}(x,y)$.

To prove the lemma, it suffices to show these two inequalities both hold:

$$\inf_{\zeta \in \Gamma(P,Q)} \text{ess sup}_{(x,y) \in (\Theta \times \Theta, \zeta)} \text{dist}(x,y) \leq \inf\{\alpha > 0 : P(U) \leq Q(U^\alpha), \forall \text{ open } U \subset \Theta\} \tag{17}$$

$$\inf_{\zeta \in \Gamma(P,Q)} \text{ess sup}_{(x,y) \in (\Theta \times \Theta, \zeta)} \text{dist}(x,y) \geq \inf\{\alpha > 0 : P(U) \leq Q(U^\alpha), \forall \text{ open } U \subset \Theta\} \tag{18}$$

For the sake of clarity, we denote

$$W_{\text{LHS}} = \inf_{\zeta \in \Gamma(P,Q)} \text{ess sup}_{(x,y) \in (\Theta \times \Theta, \zeta)} \text{dist}(x,y) = \inf_{\zeta \in \Gamma(P,Q)} \|\text{dist}(\cdot,\cdot)\|_{L_\infty(\Theta^2, \zeta)}, \text{ and}$$

$$W_{\text{RHS}} = \inf\{\alpha > 0 : P(U) \leq Q(U^\alpha), \forall \text{ open } U \subset \Theta\}.$$

We first prove (17). To prove (17), it suffices to show that for any $\alpha > W_{\text{RHS}}$, the relationship $\alpha \geq W_{\text{LHS}}$ inherently holds.

In particular, we leverage the following Strassen's theorem and prove that both constants $\beta$ and $\varepsilon$ can be driven to zero.

**Lemma 28** (Strassen's Theorem, Strassen (1965)). *Suppose that $(\Theta, \text{dist})$ is a separable metric space and $\alpha, \beta > 0$. Suppose the laws $P$ and $Q$ are such that, for all open sets $U \subset \Theta$,*
$$P(U) \leq Q(U^\alpha) + \beta$$
*where $U^\alpha = \{x \in \Theta : \text{dist}(x, U) \leq \alpha\}$.*

*Then for any $\varepsilon > 0$ there exist a law $\zeta$ on $\Theta \times \Theta$ with marginals P and Q, such that*
$$\zeta(\text{dist}(x,y) > \alpha + \varepsilon) \leq \beta + \varepsilon. \tag{19}$$

Take an arbitrary $\alpha > W_{\text{RHS}}$. Our goal is to prove $\alpha \geq W_{\text{LHS}}$. By the definition of infimum, we have
$$P(U) \leq Q(U^\alpha), \forall \text{ open } U \subset \Theta,$$
For arbitrary $\beta, \varepsilon$, by Strassen's theorem, (plugging in (19)) there exist $\zeta_{\beta,\varepsilon} \in \Gamma(P,Q)$ such that
$$\zeta_{\beta,\varepsilon}(\text{dist}(x,y) > \alpha + \varepsilon) \leq \beta + \varepsilon.$$
We are going to choose $\beta, \varepsilon$ to be sufficiently small (both go to zero) and take the limits using Prohorov's Theorem and Portmanteau theorem.

Taking $\beta = \varepsilon = \frac{1}{n}$, by Strassen's theorem, there exists $\zeta_n \in \Gamma(P,Q)$ such that
$$\zeta_n\left(\text{dist}(x,y) > \alpha + \frac{1}{n}\right) \leq \frac{2}{n}.$$

Since $P, Q$ are tight, for any $\epsilon > 0$, there exists two compact sets $K_1, K_2 \subset \Theta$, such that

$$P(\Theta \setminus K_1) \le \epsilon, \ Q(\Theta \setminus K_2) \le \epsilon$$

Note that $\zeta_n \in \Gamma(P, Q)$, we have (since $\mathbf{1}\{(x, y) \notin K_1 \times K_2\} \le \mathbf{1}\{x \notin K_1\} + \mathbf{1}\{y \notin K_2\}$)

$$\zeta_n \left( (\Theta \times \Theta) \setminus (K_1 \times K_2) \right) \le \zeta_n \left( (\Theta \setminus K_1) \times \Theta \right) + \zeta_n \left( \Theta \times (\Theta \setminus K_2) \right) \le 2\epsilon$$

thus $\{\zeta_n\}_{n=1}^{\infty}$ is a tight sequence of probability measures.

Since $\{\zeta_n\}_{n=1}^{\infty}$ is tight, by Prohorov's Theorem, there exists a weakly convergent subsequence $\zeta_{n(k)} \Rightarrow \zeta$. We have $\zeta \in \Gamma(P, Q)$ because $\zeta(A \times \Theta) = \lim_{k \to \infty} \zeta_{n(k)}(A \times \Theta) = P(A)$, and that $\zeta(\Theta \times A) = \lim_{k \to \infty} \zeta_{n(k)}(\Theta \times A) = Q(A)$.

For arbitrary $\delta > 0$, the set $\{(x, y) \in \Theta \times \Theta : \text{dist}(x, y) > \alpha + \delta\}$ is a open set in $\Theta \times \Theta$. By the Portmanteau Theorem,

$$\zeta \left( \text{dist}(x, y) > \alpha + \delta \right) \le \liminf_{k \to \infty} \zeta_{n(k)} \left( \text{dist}(x, y) > \alpha + \delta \right)$$

$$\le \liminf_{k \to \infty} \zeta_{n(k)} \left( \text{dist}(x, y) > \alpha + \frac{1}{n(k)} \right)$$

$$\le \liminf_{k \to \infty} \left( \frac{2}{n(k)} \right) = 0,$$

where the first inequality is by Portmanteau Theorem. The second inequality holds because there exist $k_0$ such that $\delta > \frac{1}{n(k_0)}$. The third inequality holds due to the construction of $\zeta_n$.

Since $\delta$ is arbitrary, (taking $\delta = \frac{1}{n}$ and by Fatou's Lemma,) we have that

$$\zeta \left( \text{dist}(x, y) > \alpha \right) = 0. \tag{20}$$

Therefore, $\alpha \ge \text{ess sup}_{(x,y) \in (\Theta \times \Theta, \zeta)} \text{dist}(x, y) = W_{\text{LHS}}$. Thus (17) is proved.

Next, we show that (18) holds. Similarly, to prove (17), it suffices to show that for any $t > W_{\text{LHS}}$, the relationship $t \ge W_{\text{RHS}}$ inherently holds.

Take an arbitrary $t > W_{\text{LHS}}$, our goal is then to prove that $t \ge W_{\text{RHS}}$. By the definition of infimum, there exists a $\zeta \in \Gamma(P, Q)$, such that

$$\underset{(x,y) \in (\Theta \times \Theta, \zeta)}{\text{ess sup}} \ \text{dist}(x, y) < t.$$

Therefore, $\zeta \left( \{(x, y) \in \Theta^2 : \text{dist}(x, y) > t\} \right) = 0$, which translate into

$$\int \mathbf{1}\{\text{dist}(x, y) > t\} \mathrm{d}\zeta(x, y) = 0. \tag{21}$$

Thus, we obtain that

$$Q(U^t) = Q\left(\{y \in \Theta : \mathrm{dist}(y, U) \leq t\}\right)$$

$$= \int \mathbf{1}\{x \in \Theta\}\mathbf{1}\{\mathrm{dist}(y, U) \leq t\}\mathrm{d}\zeta(x, y)$$

$$\geq \int \mathbf{1}\{x \in U\}\mathbf{1}\{\mathrm{dist}(y, U) \leq t\}\mathrm{d}\zeta(x, y)$$

$$= \int \mathbf{1}\{x \in U\}\mathbf{1}\{y \in \Theta\}\mathrm{d}\zeta(x, y) - \int \mathbf{1}\{x \in U\}\mathbf{1}\{\mathrm{dist}(y, U) > t\}\mathrm{d}\zeta(x, y)$$

$$\text{(If } x \in U \text{ and } \mathrm{dist}(y, U) > t, \text{ then } \mathrm{dist}(x, y) > t.)$$

$$\geq \int \mathbf{1}\{x \in U\}\mathbf{1}\{y \in \Theta\}\mathrm{d}\zeta(x, y) - \int \mathbf{1}\{\mathrm{dist}(x, y) > t\}\mathrm{d}\zeta(x, y) \qquad \text{(Plug in (21))}$$

$$= \int \mathbf{1}\{x \in U\}\mathbf{1}\{y \in \Theta\}\mathrm{d}\zeta(x, y)$$

$$= P(U).$$

Since $Q(U^t) \geq P(U)$, we have

$$t \in \{r > 0 : P(U) \leq Q(U^r), \forall \text{ open } U \subset \Theta\}.$$

Thus

$$t \geq \inf\{\alpha > 0 : P(U) \leq Q(U^\alpha), \forall \text{ open } U \subset \Theta\} = W_{\mathrm{RHS}}.$$

Therefore, (18) holds.

Since (17) and (18) hold, we have

$$\inf_{\zeta \in \Gamma(P,Q)} \operatorname*{ess\,sup}_{(x,y) \in (\Theta \times \Theta, \zeta)} \mathrm{dist}(x, y) = \inf\{\alpha > 0 : P(U) \leq Q(U^\alpha), \forall \text{ open } U \subset \Theta\},$$

which demonstrates the equivalence of the two definitions.

We now prove the attainability of the infimum in Definition 7 by repeating the above construction of $\zeta$ in the first part of our proof. We provide a simplified proof as follows.

The above construction of $\zeta$ before (20) tells us that for an arbitrary $\alpha > W_\infty(P, Q)$, there exists a $\zeta \in \Gamma(P, Q)$ such that $\zeta\left(\mathrm{dist}(x, y) > \alpha\right) = 0$. Taking $\alpha_m = W_\infty(P, Q) + \frac{1}{m}$, there exists a sequence $\{\zeta_m\} \subset \Gamma(P, Q)$ such that $\zeta_m\left(\mathrm{dist}(x, y) > \alpha + \frac{1}{m}\right) = 0$. Since $\{\zeta_m\} \subset \Gamma(P, Q)$, likewise, we obtain that $\{\zeta_m\}$ is tight. By Prohorov's Theorem, there exists a weakly convergent subsequence $\zeta_{m(k)} \Rightarrow \zeta^*$. Similarly, $\zeta^* \in \Gamma(P, Q)$. By the same token, for arbitrary $\delta > 0$, we have

$$\zeta^*\left(\mathrm{dist}(x, y) > W_\infty(P, Q) + \delta\right) \leq \liminf_{k \to \infty} \zeta_{m(k)}\left(\mathrm{dist}(x, y) > W_\infty(P, Q) + \delta\right)$$

$$\leq \liminf_{k \to \infty} \zeta_{m(k)}\left(\mathrm{dist}(x, y) > W_\infty(P, Q) + \frac{1}{m(k)}\right)$$

$$= 0,$$

where the first inequality is by the Portmanteau Theorem. The second inequality holds because there exist $k_0$ such that $\delta > \frac{1}{m(k_0)}$. The third inequality holds due to the construction of $\zeta_m$.

Since $\delta$ is arbitrary, (taking $\delta = \frac{1}{n}$ and by Fatou's Lemma,) we have that

$$\zeta^*\left(\mathrm{dist}(x, y) > W_\infty(P, Q)\right) = 0.$$

That is,

$$\zeta^* \left(\text{dist}(x,y) \leq W_\infty(P,Q)\right) = 1.$$

Therefore,

$$W_\infty(P,Q) \geq \underset{(x,y)\in(\Theta\times\Theta,\zeta^*)}{\text{ess sup}} \text{dist}(x,y).$$

On the other hand, by the definition of $W_\infty$, since $\zeta^* \in \Gamma(P,Q)$, we have

$$W_\infty(P,Q) \leq \inf_{\zeta\in\Gamma(P,Q)} \underset{(x,y)\in(\Theta\times\Theta,\zeta)}{\text{ess sup}} \text{dist}(x,y) \leq \underset{(x,y)\in(\Theta\times\Theta,\zeta^*)}{\text{ess sup}} \text{dist}(x,y).$$

Therefore, $W_\infty(P,Q) = \text{ess sup}_{(x,y)\in(\Theta\times\Theta,\zeta^*)} \text{dist}(x,y)$, for $\zeta^* \in \Gamma(P,Q)$, which proves the attainability of the infimum in the definition of $W_\infty$. ∎

## J  FACTS ABOUT NOISY GRADIENT DESCENT

Noisy gradient descent is an alternative algorithm that can be used to obtain pure-DP or pure-Gaussian DP under the same assumptions we have. While it uses full batch gradients, its analysis relies on the theory of stochastic gradient descent since the full gradients in each iteration are perturbed by either the Laplace mechanism or the Gaussian mechanism. We explicitly write out the guarantees of noisy gradient descent in this section so as to substantiate our discussion related to our computational guarantee of localized-ASAP with MALA. We focus the discussion on Gaussian DP in the $\alpha n$-strongly convex setting.

**Theorem 4** (Noisy Gradient Descent for Lipschitz and strongly Convex Losses). *Assume the loss function $\ell(\theta,(x,y))$ is $G$-Lipschitz for any data point $(x,y)$. Assume $\sum_i \ell_i(\theta)$ is $\alpha n$-strongly convex on $\Theta$. Consider the following (projected) noisy gradient descent algorithm that initializes at $\theta_0$ and update the parameter by $T$ rounds using*

$$\theta_t = \text{Proj}_\Theta(\theta_{t-1} - \eta_t(\sum_{i=1}^n \nabla\ell_i(\theta_{t-1}) + \mathcal{N}(0, \frac{TG^2}{2\mu^2}I_d)))$$

*with $\eta_t = \frac{1}{\lambda t}$. Let $\theta_\lambda^*$ be the minimizer of the regularized ERM problem and $\theta^*$ to be the minimizer of the unregularized ERM problem.*

   *1. This algorithm that releases the whole trajectory $\theta_1, ..., \theta_T$ satisfies satisfies $\mu$-GDP.*

   *2. It also satisfies that*

$$\mathbb{E}\left[\mathcal{L}\left(\sum_{t=1}^T \frac{2t}{T(T+1)}\theta_t\right)\right] - \mathcal{L}(\theta_\lambda^*) \leq \frac{4n^2G^2}{\alpha n(T+1)} + \frac{dG^2}{2\alpha n\mu^2}.$$

   *3. If $T \geq \frac{8n^2\mu^2}{d}$, then it achieves an excess empirical risk of $\frac{dG^2}{\alpha n\mu^2}$.*

**Proof** Observe that the $G$-Lipschitz loss says that the global sensitivity of the gradient is $G$. The privacy analysis follows from the composition of Gaussian mechanism via Gaussian DP for $T$ rounds (Dong et al., 2022). This releases the entire sequence of parameters. The weighted average of the parameters over time is post-processing. The second statement follows from Section 3.2 of (Lacoste-Julien et al., 2012) by choosing the noise level and other parameters appropriately. The last statement is corollary by choosing $T$ to be large. ∎

In the above, the excess empirical risk achieves the optimal rates but requires the algorithm to run for $O(n^2)$ iterations. In the smooth case, one can obtain faster convergence but at the cost of the resulting excess empirical risk.

**Theorem 5** (Noisy Gradient Descent for smooth and strongly convex Losses). *Consider the same algorithm and assume all assumptions from Theorem 4. In addition, assume individual loss functions $\ell_x$ are $\beta$-smooth.*

1. *This algorithm that releases the whole trajectory $\theta_1, ..., \theta_T$ satisfies satisfies $\mu$-GDP.*

2. *Choose a constant learning rate $\eta \leq \frac{1}{n\beta}$, run for $T$ iterations*

$$\mathbb{E}\left[\mathcal{L}\left(\theta_T\right)\right] - \mathcal{L}(\theta_\lambda^*) \leq \frac{n\beta}{2}(1 - \eta\alpha n)^T \mathrm{Diam}(\Theta)^2 + \frac{\eta n \beta d T G^2}{n\alpha\mu^2}$$

3. *Choose $\eta = 1/(n\beta)$ and $T = O(\frac{\beta}{\alpha}\log n)$, the excess risk is*

$$\mathbb{E}\left[\mathcal{L}\left(\theta_T\right)\right] - \mathcal{L}(\theta_\lambda^*) \leq O(\frac{\beta d G^2 \log n}{n\alpha^2 \mu^2}).$$

**Proof** From Theorem 5.7 of (Garrigos & Gower, 2023) we can derive the convergence in various regimes. ∎

It is not hard to see that there isn't a good choice of the learning rate parameter $\eta$ based on the bound above that can get rid of the additional $\frac{\beta \log n}{\alpha}$ factor.

