# OpenReview forum: "Tractable MCMC for Private Learning with Pure and Gaussian Differential Privacy"
_ICLR.cc/2024/Conference — ICLR 2024 poster_

### Official Review · Reviewer_qRJd · 2023-10-17

**Soundness:** 4 excellent
**Presentation:** 3 good
**Contribution:** 3 good
**Rating:** 8
**Confidence:** 3

**Summary:**

The paper proposes an algorithm for pure DP or GDP DP-ERM that achieves nearly linear time complexity with an optimal error rate under regularity assumptions on the loss. The algorithm uses the exponential mechanism with MCMC sampling, and uses a novel analysis to show that the approximate sample from the MCMC can still meet pure DP or GDP with some additional privacy loss.

**Strengths:**

The paper is fairly easy to understand, despite having lot's of theory. The MCMC technique of obtaining a pure DP or GDP guarantee from an approximate exponential mechanism is novel to my knowledge, and could find applications elsewhere, although implementing the mechanism in practice doesn't currently seem possible. While the performance guarantee of the proposed DP-ERM algorithm is purely theoretical, it advances the current knowledge on what is possible under DP.

**Weaknesses:**

The biggest weakness of the paper is the purely theoretical nature of the method in its current form. Implementing it would be difficult, as the required number of MALA iterations is only given as an asymptotic expression, so it would not be possible to know the required number of iterations for any given problem. If possible, it would be great to have a non-asymptotic expression for the number of iterations in the Appendix, or some other way to compute it.

Minor comments:
- Lemma 14 should be mentioned in the proof of Theorem 2 when referring to the adaptive composition theorems. In the proof, I think it is possible that for some $y$, the sensitivity of $f_{u,y}$ is greater than the bound with more than probability 0 under $\zeta(\cdot | y) \times \zeta(\cdot | y)$, but this set of $y$ would have 0 measure under $\zeta$ or $\zeta'$. I think the proof still works due to Lemma 14 allowing the second mechanism to be almost surely DP, but this should be explicitly mentioned.
- In footnote 1, both sides of the DP inequality have $D'$.
- In the $P(A)$ expression in the GDP part of the proof of Lemma 14, line 3 is a repeat of line 2. In the expression of $H_\alpha(\mathcal{M}(D) || \mathcal{M}(D'))$, there's an extra $\frac{d P_1}{d Q_1}(x)$ in the first indicator on line 7. When that expression continues on the next page, the Gaussian Radon-Nikodym derivatives involving $\mu_2$ at the end of the integral are the wrong way around on lines 2-5.

**Questions:**

- Would it be possible to use ASAP outside ERM, with a generic utility function $F(\theta)$ for the exponential mechanism? What conditions would $F$ need to meet?
- When discussing related work, you mentioned that an existing algorithm could solve DP-SCO in nearly linear time, but adapting their algorithm to DP-ERM would not result in a linear time. What are the differences between DP-ERM and DP-SCO, and how do they lead to this difference?
- Isn't Lemma 5 effectively assuming that the loss is bounded, as Lipschitzness of the loss implies continuity, and $\Theta$ must be bounded due to the upper bound on $\gamma$?
- In the proof of Theorem 1, shouldn't the expressions for $K$ have $\Omega$, not $\mathcal{O}$, so they would be lower bounds instead of upper bounds?

---

> ### Author Response · Authors · 2023-11-16
>
> >1. Would it be possible to use ASAP outside ERM, with a generic utility function $F(\theta)$ for the exponential mechanism? What conditions would need to meet?
>
> Yes, our results can be extended to non-convex settings by incorporating techniques from [Ma et al. 2019]. The results of our paper also directly apply to the utility function $F$ with smoothness and strong convexity.
>
> >2. When discussing related work, you mentioned that an existing algorithm could solve DP-SCO in nearly linear time, but adapting their algorithm to DP-ERM would not result in a linear time. What are the differences between DP-ERM and DP-SCO, and how do they lead to this difference?
>
> Thanks for pointing it out. To enhance clarity, we rephrase the discussion on Page 9 as follows: "To avoid any confusion, a nearly linear-time algorithm for DP-SCO in the smooth/strongly convex setting **that achieves optimal rates** exists (Feldman et al., 2020). But to the best of our knowledge, the problem remains open for DP-ERM until this paper." DP-ERM and DP-SCO have different optimal rates.  For the strongly convex case, the DP-SCO has $\frac{G^2}{\alpha}(\frac{1}{n} + \frac{d^2}{n^2\epsilon^2})$ for $\epsilon$-DP or $\frac{G^2}{\alpha}(\frac{1}{n} + \frac{d}{n^2\mu^2})$ for $\mu$-GDP while DP-ERM's optimal rates are $\frac{d^2}{n^2\epsilon^2}$ and $\frac{d}{n^2\mu^2}$.  The latter is harder to achieve in the constant $d$, large $n$ regime.  Existing near-linear-time DP-SCO algorithms with optimal rates (only under zCDP and approx-DP --- no such results for pure-DP or pure-Gaussian methods exist) mostly leverage the fact that  $\frac{G^2}{\alpha n}$ is bigger and one can stop versions of DP-GD or DP-SGD training early.
>
> Regarding DP-SCO vs DP-ERM: We love both settings. DP-ERM is a bit stronger and cleaner due to the separation between optimization and generalization. ERM and its generalization bound and adaptivity (e.g., low-noise, fast spectral decay, stability) have been studied for much longer than DP-SCO. DP-SCO algorithms focus on weakening computation so as to just achieve the worst-case excess (population) risk for crude classes of problems but do not adapt to finer subclasses. DP-ERM however automatically achieves whatever adaptivity and other properties that ERM enjoys. This is a concrete benefit of DP-ERM over DP-SCO in our minds.
>
> >3. Isn't Lemma 5 effectively assuming that the loss is bounded, as Lipschitzness of the loss implies continuity, and $\Theta$ must be bounded due to the upper bound on $\gamma$?
>
> The key insight from Lemma 5 is that it avoids assuming a bound for $F$, which is potentially large. The loss may not be bounded actually unless we assume there is a point $\theta\in\Theta$ such that the loss $F(\theta)$ is bounded. For example, let's say the loss is $\log(1+e^{-y (\theta\cdot x)})$, with $\mathcal{X}=[-1,1]$, $\Theta = [10000,10001]$. Then if $y=-1,x=1$, the loss can be as large as  $10001$.  This is a vacuous bound compared to $G \cdot \mathrm{Diam}(\Theta) \leq 1$.   Compared to the classical analysis based on the bounded loss, the bounded-range analysis is both cleaner and tighter.
>
> >4. In the proof of Theorem 1, shouldn't the expressions for
> $K$ have $\Omega$, not $\mathcal{O}$, so they would be lower bounds instead of upper bounds?
>
> Yes. They should be lower bounds. Thanks for pointing it out.
>
> >5. The biggest weakness of the paper is the purely theoretical nature of the method in its current form. Implementing it would be difficult, as the required number of MALA iterations is only given as an asymptotic expression, so it would not be possible to know the required number of iterations for any given problem. If possible, it would be great to have a non-asymptotic expression for the number of iterations in the Appendix, or some other way to compute it.
>
> Thank you for pointing it out. We omit the constants in our paper for simplicity and will discuss them in the appendix in the upcoming revision. The constant is within 500 and can be found in the proof of Lemma 7 of [Ma et al. 2019], which is Lemma 9 in the arxiv version.
>
> Reference:
>
> Ma, Y. A., Chen, Y., Jin, C., Flammarion, N., & Jordan, M. I. (2019). Sampling can be faster than optimization. Proceedings of the National Academy of Sciences, 116(42), 20881-20885.

---

> > ### Comment · Reviewer_qRJd · 2023-11-16
> >
> > Thanks for the reply. You addressed my main points, especially if you get the experiments that you mentioned in the other reply done during the discussion period update the submission to include them.
> >
> > > The key insight from Lemma 5 is that it avoids assuming a bound for $F$, which is potentially large.
> >
> > So the key benefit is not needing to use a potentially large bound on $F$ in the privacy calculations? I would explicitly mention that as a benefit of Lemma 5 if that is the case.

---

> > > ### Author Response · Authors · 2023-11-17
> > >
> > > Yes, we will explicitly mention this in our revision. Thank you for your suggestions.

---

### Official Review · Reviewer_kgTq · 2023-10-30

**Soundness:** 3 good
**Presentation:** 3 good
**Contribution:** 3 good
**Rating:** 6
**Confidence:** 3

**Summary:**

The authors propose a novel MCMC algorithm for pure DP and Gaussian DP empirical risk minimization. The new MCMC sampling scheme does not introduce extra privacy failure probablity and thus preserves pure DP. Moreover, the algorithm runs nearly linearly in the dataset size.

**Strengths:**

1. DP-ERM is a very fundamental task. The paper addresses very practical and important issues in differentially private optimization.
2. The paper proposes a new algorithm for DP-ERM. The major benefit is that it avoids catastrophic privacy failures and thus guarantees pure DP, and runs linearly with the dataset size $n$.
3. The paper has interesting technical contributions, for example, a new MCMC algorithm, the relation between TV distance and  Wasserstein-infinity distance.

**Weaknesses:**

1. $\theta^*$ seems to be defined as the minimizer for the regularized loss in Section 2.1, but in Assumption 3 $\theta^*$ is used again for the minimizer of the original loss. The authors should clarify what is the $\theta^*$ in Theorem 3. This is a crucial problem as in existing DP-ERM literature (e.g. Bassily'14) we should care about the excess risk w.r.t. $\min_{\theta}\mathcal{L}(\theta)$, not the regularized loss. If $\theta^*$ is the minimizer of the regularized loss, then the authors should clarify why it is used instead of the commonly adopted formulation of excess risk.
2. In view of 1 the notation causes confusion in understanding the main theoretical results. There are other notation issues: $J$ was first defined as $\sum_{i}\ell_i$ which collides with $\mathcal{L:}$, and then redefined in Algorithm 3 as the regularized loss centered at $\theta_0$. The authors should check other notation issues.
3. The algorithm requires rejection sampling which in the worst case may not terminate.
4. A minor issue is that the work lacks empirical evaluations. Even a small experiment on even the simplest convex optimization problem can provide very strong support for the computation efficiency of this work and demonstrate the practicality of the proposed algorithm.

====Update====
Raised my score to 6 after the authors have addressed my concerns.

**Questions:**

1. See weakness 1.
2. If in Algorithm 1, we are only allowed to perform the sampling N times, and thus must return a FAIL state if none of the $\theta^K$s are valid, would the algorithm still be pure-DP or pure Gaussian DP?

---

> ### Author Response · Authors · 2023-11-16
>
> >1. $\theta^*$ seems to be defined as the minimizer for the regularized loss in Section 2.1, but in Assumption 3 $\theta^*$ is used again for the minimizer of the original loss. The authors should clarify what is the $\theta^*$ in Theorem 3. This is a crucial problem as in existing DP-ERM literature (e.g. Bassily'14) we should care about the excess risk w.r.t. $\min_{\theta}\mathcal{L}(\theta)$, not the regularized loss. If $\theta^*$ is the minimizer of the regularized loss, then the authors should clarify why it is used instead of the commonly adopted formulation of excess risk.
>
> In this paper, we only consider the excess risk without the regularizer, i.e., $\lambda=0$. We set $\lambda=0$ in the first line of the input of Algorithm 3, as well as in Table 2. In this context, the notations align, with $J=\mathcal{L}=\sum_i \ell_i$, and $\theta^*=\textrm{argmin}_{\theta}\mathcal{L}(\theta)$. We keep the notation of regularizer to allow more flexibility for our algorithm and future work.
>
> >2. In view of 1 the notation causes confusion in understanding the main theoretical results. There are other notation issues: $J$ was first defined as $\sum_{i}\ell_i$
>  which collides with $\mathcal{L}$, and then redefined in Algorithm 3 as the regularized loss centered at $\theta_0$. The authors should check other notation issues.
>
> See the response of 1. Thank you for your suggestions; we will check and revise accordingly.
>
> >3. If in Algorithm 1, we are only allowed to perform the sampling N times, and thus must return a FAIL state if none of the $\theta^K$s are valid, would the algorithm still be pure-DP or pure Gaussian DP?
>
> We did not include the case when the sampler can fail in the submitted version for the sake of simplicity. As a consequence, as you alluded to there is a very small chance that the algorithm does not halt within a short time. What you proposed can be combined quite seamlessly with the ASAP technique that we proposed in this paper to obtain a worst-case run-time bound by stopping after $N$ attempts and outputting an arbitrary point from the domain $\Theta$. Let's say the chance of failure after $N$ attempts is $\delta$, then we can essentially just add $\delta$ to our $\xi$ parameter so we get a TV-distance-bound of $\delta+\xi$ before converting it to $W_\infty$ and use ASAP to obtain a pure-DP mechanism. Details are as follows:
>
> Let $X$ be the "SUCCESS or FAIL" random variable with $\mathbb{P}(X=1)=1-\delta$, and $\mathbb{P}(X=0)=\delta$. Denote $p_S$ as the distribution of output in a successful case, and denote $p_F$ as an arbitrary distribution within domain $\Theta$. The output $\tilde{\theta}$ of Algorithm 1 (with failure output) follows:
> $$\tilde{\theta}|X=1 \sim p_S, \textrm{ and } \tilde{\theta}|X=0 \sim p_F.$$
> Denote $\tilde{p}$ as the distribution of $\tilde{\theta}$. Then we have $\tilde{p}=(1-\delta)\cdot p_S+\delta \cdot p_F$. Thus
> $$d_{TV}(\tilde{p},p_S)=\sup_{A}|\tilde{p}(A)-p_S(A)|=\sup_{A}|(1-\delta)p_S(A)+\delta p_F(A)-p_S(A)|\leq \delta.$$
> Denote $p^*$ as the reference distribution. Since $d_{TV}(p_S, p^*)\leq \xi$, we thereby have
> $$d_{TV}(\tilde{p}, p^*)\leq d_{TV}(\tilde{p}, p_S)+d_{TV}(p_S, p^*)\leq \delta+\xi.$$
> Replacing $\xi$ by $\delta+\xi$ in Theorem 1 implies the pure DP of the algorithm.
>
> >4. The algorithm requires rejection sampling which in the worst case may not terminate.
>
> We demonstrate in our analyses that the algorithm will terminate with high probability $1-q$ in $$\Omega( d (\ln \kappa+\kappa \ln(d/\rho) + \ln n ) \max\lbrace \kappa^{3/2}\sqrt{d(\kappa \ln(d/\rho) + \ln n )}, d\kappa \rbrace \ln(1/q) )$$ steps. You are correct; there is an exponentially small chance it will take longer, but the probability diminishes with the amount of runtime. Moreover, as clarified in our response to Question 3, our algorithm maintains the pure DP guarantee even if we need to terminate our algorithm in N steps.
>
> >5. A minor issue is that the work lacks empirical evaluations. Even a small experiment on even the simplest convex optimization problem can provide very strong support for the computation efficiency of this work and demonstrate the practicality of the proposed algorithm.
>
> Thank you for your suggestions. Our next revision of the paper will likely include some experiments. We should also note that "Private Convex Optimization via Exponential Mechanism" by Gopi et al. doesn't have any experiments, nor do many existing works on sampling. So while we agree on the importance of including experiments to demonstrate practicality, we want to point out that there is also precedent for theory-only work.
>
> Reference:
>
> Gopi, S., Lee, Y. T., & Liu, D. (2022). Private convex optimization via exponential mechanism. In Conference on Learning Theory, pages 1948-1989. PMLR.

---

> ### Author Response · Authors · 2023-11-21
> **Appreciation for Your Review and Further Evaluation**
>
> We deeply appreciate your invaluable review. In our updated manuscript and response, we have dedicated ourselves to thoroughly addressing all your concerns and questions.
>
> Our revision incorporates experiments and clarifies the notation of $\theta^*$ in Section 2.1. Additionally, our response emphasizes that our algorithm maintains pure DP or pure Gaussian DP guarantee, *even in the presence of a potential FAIL case*.
>
> Could you please spare a moment to review our response? We are eager for your feedback and prepared to provide any further clarifications.
>
> Thank you for your time and consideration.

---

> > ### Comment · Reviewer_kgTq · 2023-11-21
> > **Thank you for your response.**
> >
> > Thank you for your response. I think you have addressed my concerns. I am willing to raise my score to 6 if I don't find other major issues.

---

### Official Review · Reviewer_kWWS · 2023-11-01

**Soundness:** 3 good
**Presentation:** 3 good
**Contribution:** 3 good
**Rating:** 6
**Confidence:** 4

**Summary:**

The paper studies the problem of approximate posterior sampling with pure differential privacy guarantee, for which prior works either have only proved weaker notions of DP guarantees (such as approximate DP and R\'enyi DP) or have only proposed computationally expensive pure DP algorithms. The difficulty in establishing pure differential privacy guarantees lies in the gap between approximate and exact posterior sampling, contributing to a non-zero probability of unbounded information leakage. The authors circumvent this limitation by proving that as long as the posterior sampling process converges in $W_{\infty}$ distance to a reference distribution that satisfies pure DP, then perturbing the MCMC sample with calibrated noise also satisfies pure DP. For strongly convex smooth loss functions, the author then designed a posterior sampling algorithm that converges in $W_{\infty}$ distance by combining the Metropolic-Hasting algorithm (that converges in total variation distance) and a rejection step that only accepts samples within a bounded ball (this boundedness enables conversion from total variation distance bound to $W_{\infty}$ distance bound). Finally, the authors showcased an application of their DP posterior sampling algorithm for the DP empirical risk minimization problem, which achieves the optimal rates in near-linear runtime.

**Strengths:**

- The idea of perturbing an MCMC sampling process (that only satisfies weaker notions of DP guarantees) with noise to achieve the stronger pure-DP guarantee is interesting and novel. To make this idea concrete, the paper has used an interesting new technique that converts the weaker  TV distance bound to a $W_{\infty}$ distance bound under the conditions that the domain is a bounded ball and the posterior density is bounded away from zero.

- The proposed algorithm achieves a near-optimal rate within near-linear runtime under pure DP for the DP ERM problem, under strongly convex smooth loss function on $\ell_2$-bounded ball). The proved rate improves with a multiplicative factor $\kappa\log n$ where $\kappa$ is the condition number, and $n$ is the number of data records.

**Weaknesses:**

- The improvement for DP ERM due to the designed approximate posterior sampling algorithm (and several parts of the proofs) requires further clarification. See questions 1 and 2 for more details.

- The setting is quite constrained, i.e., strongly convex and smooth loss functions over $\ell_2$-bounded ball. It is unclear whether the privacy analysis, especially the critical conversion theorem (from total variation distance bound to $W_{\infty}$ distance bound), is generalizable to more general forms of bounded domain or loss functions.

**Questions:**

1. In the proof for Theorem 3 (improved utility bound for DP-ERM), equations (5) and (6) utilize Lemma 6 to analyze the **approximate** posterior sampling component MALA with constraint. However, Lemma 6 only holds under **exact** posterior sampling from $p(\theta)\propto \exp(-\gamma F(\theta))$. Could the authors explain this discrepancy and justify this usage of Lemma 6?

2. In proof of Theorem 1, when proving iteration complexity guarantee for convergence of MALA in total variation distance, the authors cited [Dwivedi et al. 2019] and [Ma et al. 2019]. Could the authors cite the specific theorems? Otherwise, it is hard to validate these claims.

3. Could the authors elaborate on the possibility of extending the privacy bound to more general settings, such as general bounded domain (e.g., probability simplex) or more relaxed convex/non-convex/non-smooth loss functions?

Minor comments:
- The idea of perturbing distributions with bounded $W_{\infty}$ distance with noise to strengthen the differential privacy guarantee seems quite similar to the shift reduction lemma in the "privacy amplification by iteration" analysis [Lemma 20, a]. The main difference seems to be that [Lemma 20, a] focuses on Rényi DP guarantees while the authors focus on pure DP and Gaussian DP guarantees. Maybe the authors could add more discussion regarding the non-trivialness of this extension.

Reference:
- [a] Feldman, Vitaly, Ilya Mironov, Kunal Talwar, and Abhradeep Thakurta. "Privacy Amplification by Iteration." arXiv preprint arXiv:1808.06651 (2018).

---

> ### Author Response · Authors · 2023-11-17
>
> >1. In the proof for Theorem 3, equations (5) and (6) utilize Lemma 6 to analyze the approximate posterior sampling component MALA with constraint. However, Lemma 6 only holds under exact posterior sampling from $p(\theta)\propto \exp(-\gamma F(\theta))$. Could the authors explain this discrepancy and justify this usage of Lemma 6?
>
> Thank you for bringing up this point. We omitted the analysis regarding the discrepancy between the approximate posterior and the exact posterior for the sake of simplicity. This simplification doesn't compromise the results, as this discrepancy is **dominated** by the error arising from the noise added in the final step of ASAP, as shown in (4). To clarify why this simplification does not impact the established bound in Theorem 3, we present a detailed analysis below.
>
> Take $\Delta=\frac{d G \log(d/\rho)}{2 n^2 \alpha \varepsilon}$ as Table 2. Denote $\tilde{\theta}\sim \tilde{p}$ the output of MALA with constraint, and denote $p^*$ the exact posterior. Since $W_\infty(\tilde{p},p^*)\leq \Delta$, applying Lemma 21, there exist a coupling of $\tilde{p}$ and $p^*$, denoted as $\zeta$, such that $esssup_{(x,y)\in (\Theta\times\Theta, \zeta) }\lVert x-y \rVert_2\leq \Delta$. Take $\theta_{exact}| \tilde{\theta}\sim \zeta(\cdot|\tilde{\theta})$, then we know $\theta_{exact}\sim p^*$.
> With the $nG$-Lipschitz continuity of $\mathcal{L}$, this discrepancy is bounded by
> $$\mathbb{E} _ {(\tilde{\theta},\theta _ {exact})\sim \zeta}[\mathcal{L}(\tilde{\theta})-\mathcal{L}(\theta _ {exact})]\leq n G \mathbb{E} _ {(\tilde{\theta},\theta_{exact})\sim \zeta}[\lVert\tilde{\theta}-\theta _ {exact}\rVert_2]\leq n G \Delta.$$
> This discrepancy is then dominated by the upper bound for $\mathbb{E}[\mathcal{L}(\hat{\theta})-\mathcal{L}(\tilde{
> \theta})]$, which is $\frac{\sqrt{2 d}n G \Delta}{\varepsilon}$ as shown in (4).
> Therefore, the discrepancy between the approximate posterior and the exact posterior has a negligible impact on the empirical risk bound.
>
> Thanks again for bringing up this point; We will add a more detailed analysis to our upcoming revision.
>
> >2. In proof of Theorem 1, when proving iteration complexity guarantee for convergence of MALA in total variation distance, the authors cited [Dwivedi et al. 2019] and [Ma et al. 2019]. Could the authors cite the specific theorems?
>
> We cited Theorem 1 of [Dwivedi et al. 2019], and Lemma 8 of [Ma et al. 2019], which is in the appendix. Lemma 8 of [Ma et al. 2019] corresponds to Lemma 10 in the Arxiv version.
>
> >3. Could the authors elaborate on the possibility of extending the privacy bound to more general settings, such as general bounded domain (e.g., probability simplex) or more relaxed convex/non-convex/non-smooth loss functions?
>
> Yes, as addressed in the response to qRJd, our results can be extended to non-convex settings by leveraging techniques from [Ma et al. 2019].
>
> We would clarify that our parameter space is not necessarily an $\ell_2$ ball. Rather, We first localize it to an $\ell_2$ ball, and then perform ASAP. With regard to the probability simplex example, we can use a mirror map towards the Euclidean space, perform localized-ASAP on the Euclidean space, and then map the output back to the simplex.
>
> >4. The idea of perturbing distributions with bounded $W_\infty$ distance with noise to strengthen the differential privacy guarantee seems quite similar to the shift reduction lemma in the "privacy amplification by iteration" analysis [Lemma 20, a]. The main difference seems to be that [Lemma 20, a] focuses on Rényi DP guarantees while the authors focus on pure DP and Gaussian DP guarantees. Maybe the authors could add more discussion regarding the non-trivialness of this extension.
>
> Thanks for pointing out this interesting technical connection. As you said, the results are different. They focused on Renyi DP while we stated results for pure DP and Gaussian DP in Theorem 2. The proof techniques are also quite different. We leveraged Lemma 15, a measure-theoretic adaptive composition theorem of privacy profiles (Hockey-Stick divergences or its dual representation — $f$-DP) that we proved in this paper, which allows the exclusion of a measure 0 set.  This is a non-trivial technical step towards our relatively clean proof of Theorem 2. In fact, if we apply adaptive composition theorems of RDP, then we get an RDP bound for ASAP right away.  On the other hand, it is well-known that RDP bounds does not imply tight $f$-DP bound even for the Gaussian mechanism. Thanks again for your suggestion;  we will add the above discussion in our revision.
>
> References:
>
> Feldman, V., Mironov, I., Talwar, K., & Thakurta, A. (2018, October). Privacy amplification by iteration. In 2018 IEEE 59th Annual Symposium on Foundations of Computer Science (FOCS) (pp. 521-532). IEEE.
>
> Ma, Y. A., Chen, Y., Jin, C., Flammarion, N., & Jordan, M. I. (2019). Sampling can be faster than optimization. Proceedings of the National Academy of Sciences, 116(42), 20881-20885.

---

> > ### Comment · Reviewer_kWWS · 2023-11-20
> >
> > Thanks to the authors for the response. My primary concerns regarding clarity are addressed, and I have raised the score. However, the analysis still seems limited to me, as it necessitates a localization step, which seems challenging for non-convex problems. Additionally, the algorithm may return "Fail" or incur infinite running time with a small probability (as reviewer kgTq pointed out). This seems an important limitation, given that most runtime analyses for prior DP-ERM algorithms are non-probabilistic.

---

> ### Author Response · Authors · 2023-11-22
>
> Thank you for your invaluable review. To extend our work to a non-convex setting, for the sampling part, we can leverage techniques from [Ma et al. 2019]; regarding the localization step, we acknowledge the necessity for a more thorough discussion. We would also like to highlight that our algorithm, with the localization step, can be extended to the general convex case. Additionally, as clarified in our response to reviewer kgTq's question 3, our algorithm maintains the pure DP and pure Gaussian DP guarantee, even in the presence of a potential FAIL case.
>
> Thanks again for your insightful feedback.
>
> Reference:
>
> Ma, Y. A., Chen, Y., Jin, C., Flammarion, N., & Jordan, M. I. (2019). Sampling can be faster than optimization. Proceedings of the National Academy of Sciences, 116(42), 20881-20885.

---

### Meta-Review · Area_Chair_BNmp · 2023-12-14

**Metareview:**

The paper addresses the problem of approximate posterior sampling with pure differential privacy guarantee. This is a well-known approach for solving DP-ERM (and DP-SCO) problems which however previously either only approximate DP guarantees or required computationally expensive sampling. For the (rather restrictive) smooth and strongly convex case this work gives a posterior sampler that achieves pure DP and solves DP-ERM with asymptotically optimal rates in near-linear runtime. The result relies on localization and additional perturbation to convert $W_\infty$ distance to a divergence (in a way similar to privacy amplification by iteration and its application to Langevin analysis by Altschuler and Talwar, 2023). Overall, this is an interesting new result about one of the basic problems in DP convex optimization and the tools might find additional uses.

**Justification For Why Not Higher Score:**

The smooth and strongly convex seting is rather restrictive and the authors appear unaware that some of the same ideas have been already used in this context.

**Justification For Why Not Lower Score:**

see above

---

### Decision · Program_Chairs · 2024-01-16

Accept (poster)